

**Effects of 20-100 nanometre particles on liquid clouds in the clean**
**summertime Arctic**
W. R. Leaitch, A. Korolev, A. A. Aliabadi
Environment Canada, Toronto, Canada
J. Burkart, M. Willis, J.P.D. Abbatt
Department of Chemistry, University of Toronto, Toronto, Canada
H. Bozem, P. Hoor
Institute of Atmospheric Physics, University of Mainz, Mainz, Germany
F. Köllner, J. Schneider
Particle Chemistry Department, Max Planck Institute for Chemistry, Mainz, Germany
A. Herber, C. Konrad
Alfred Wegener Institute for Polar and Marine Research, Bremerhaven, Germany
R. Brauner
Jade University, Elsfleth, Germany
Date: January 27, 2016
Correspondence to Richard.Leaitch@Canada.ca



**Abstract.**  Aerosol-cloud research in the Arctic has largely focused on the springtime resulting
in relatively little information about the effects of the atmospheric aerosol on Arctic clouds
during summer. An airborne study, carried out during July, 2014 from Resolute Bay, Nunavut,
Canada, provides a comprehensive in-situ look into some effects of aerosol particles on liquid
clouds in the clean environment of the Arctic summer.  The median cloud droplet number
concentrations (CDNC) are 10 cm$^{-3}$ and 101 cm$^{-3}$ for low altitude cloud (LA: clouds topped
below 200 m) and higher altitude cloud (HA: clouds based above 200 m), respectively. The
mean lower activation size of aerosol particles is ≤50 nm diameter in 40% of the cases, and
particles as small as 20 nm activated in the HA clouds consistent with the higher supersaturations
inferred for those clouds. Over 60% of the LA cloud cases fall into the cloud condensation nuclei
(CCN)-limited regime of Mauritsen et al. (2011) within which increases in CDNC may increase
liquid water and warm the surface.  In that CCN-limited regime, the liquid water contents (LWC)
and the CDNC are positively correlated, but there is no dependence of changes in either the
CDNC or LWC on the aerosol, suggesting no aerosol limitation. Above the Mauritsen limit,
where indirect cooling may result, particles with diameters from 20 nm to 100 nm exert a strong
influence on the CDNC. Based on CO concentrations, the background CDNC are estimated to
range between 16 cm$^{-3}$ and 160 cm$^{-3}$, implying a large uncertainty for the baseline of the aerosol
cloud albedo effect.





## 1.    Introduction

Mass concentrations of the atmospheric aerosol in the Arctic are higher during winter and lower

during summer due to differences in the transport of the anthropogenic aerosol and wet

scavenging (e.g. Barrie, 1986; Stohl, 2006). Much of the focus of atmospheric chemistry and

aerosol-cloud research in the Arctic has been on the spring period. The transition from winter to

summer offers the opportunity to examine the changes in chemistry as the sun rises over the

polluted polar atmosphere (e.g. Barrie et al., 1988) and to study the impacts of the anthropogenic

aerosol on the Arctic solar radiation balance (e.g. Law and Stohl, 2007; Quinn et al., 2008).

Greater-than-expected warming of the Arctic (e.g. Christensen et al., 2013) and rapidly

diminishing Arctic sea ice extent (e.g. Maslanik et al., 2011) have drawn considerable attention

to the role of anthropogenic and biomass burning aerosols as warming agents for the Arctic (e.g.

Law and Stohl, 2007; Quinn et al., 2008; Shindell et al., 2008; Brock et al., 2011; Jacob et al.,

2010; UNEP, 2011; Stohl et al., 2013).  Recent evidence indicates that the net impact of the

aerosol on the Arctic over the past century has been one of cooling rather than warming (Najafi

et al., 2015).

Low-level liquid water clouds are frequent in the sunlit Arctic summer (e.g. Intrieri et al.,

2001), and these clouds can have a net cooling effect (e.g. Brenner et al., 2001; Garret et al.,

2004; Lubin and Vogelmann, 2010; Zhao and Garrett, 2015; Zamora et al., 2015;). Knowledge

of the influence of the atmospheric aerosol on climatic aspects of these clouds is complicated by

the relatively large potential differences in the albedo of the underlying surface (e.g. Herman,

1977; Lubin and Vogelmann, 2010) and the fact that the Arctic is relatively free of

anthropogenic influence in the summer, which means that aerosols from natural sources can be



the most significant sources of nuclei for cloud droplets. Those sources shift the number
distribution towards particles smaller than 100 nm (e.g. Ström et al., 2003; Engvall et al., 2008;
Tunved et al., 2013; Leaitch et al., 2013; Browse et al., 2014; Heintzenberg et al., 2015).
Particles much smaller than 100 nm are sometimes dismissed as being too small to nucleate
cloud droplets due to the assumption that the cooling mechanisms are too slow to generate the
supersaturations required to activate the smaller particles in Arctic liquid clouds (e.g. Garret et
al., 2004; Zhao and Garrett, 2015). However, the cloud supersaturation is also constrained by the
concentration of the particles larger than 100 nm. In situations with few larger particles, as in the
clean environment of the summertime Arctic, higher supersaturations can be achieved and
smaller particles may be activated (e.g. Leaitch et al., 2013).  Garrett et al. (2004) showed no
correlation of CDNC retrieved from remote sensing with surface-based total particle number
concentrations larger than about 10 nm, but total number concentration measurements do not
reflect the size distribution and the concentrations of such particles may change significantly
with altitude.  Lubin and Vogelmann (2010) estimated substantial aerosol indirect forcings for
low-level spring-summer Arctic clouds based on a "natural background range" of surface-based
total particle number concentrations of 0-50 cm$^{-3}$, which significantly underestimates the
potential contributions from natural sources as discussed in the above references.  The effect of
the background aerosol on liquid clouds has been identified as one of the most important factors
for reducing uncertainty in the aerosol cloud albedo effect (Carslaw et al., 2013). A related and
important question is how effective are particles smaller than 100 nm at nucleating cloud
droplets?

Aerosol effects on clouds may also lead to warming, but a contrast between clean versus

polluted clouds is still required (e.g. Garrett et al., 2009).  Mauritsen et al. (2011) modeled cloud



radiative forcing for low clouds using CCN number concentrations (hereafter CCNC) derived
from shipborne observations made over the Arctic Ocean (Tjernström et al., 2004; Tjernström et
al., 2014). They found that the impact from changes in CCNC for ultra-low values (<10 cm$^{-3}$),
where CCNC is equivalent to the CDNC in the model, will result in a net warming due to
associated longwave changes, whereas for CCNC>10 cm$^{-3}$ increases in the CCNC are estimated
to produce a net cooling of the atmosphere. The threshold CCNC is referred to here as the
"Mauritsen limit", and it is noted that the value of 10 cm$^{-3}$ is not considered universal.  In such
clean environments, knowledge of the natural aerosol and its influence on the microphysics of
summer clouds is critical to the assessment of the effects of the aerosol on the Arctic climate.

Past studies of Arctic aerosols and clouds have emphasized the areas of the Beaufort and

Chukchi Seas (e.g. Hobbs and Rango, 1998; Curry et al., 2001 and references therein; Lohmann
et al., 2001; Peng et al., 2002; Earle et al., 2011; Lance et al., 2011; Jouan et al., 2014; Klingebiel
et al., 2014). Most of those studies have focused on the spring when the aerosol is influenced by
anthropogenic or biomass burning aerosol. As well, there has been considerable interest in
mixed-phase clouds in the lower Arctic troposphere (e.g. Shupe et al., 2004; Sandvik et al., 2007;
Morrison et al., 2012), but a notable lack of in-situ observations of aerosols in combination with
liquid water clouds over the Arctic during summer. Among the studies that have considered in-
situ measurements of aerosols and Arctic summer clouds, Zamora et al. (2015) examined the
efficiency of biomass burning (BB) plumes on indirect forcing, estimating a forcing from these
plumes about half of the possible maximum, mostly due to the reduction in cloud-base
supersaturation by the higher concentration and larger particles that control water uptake. Shupe
et al. (2013) discussed some differences among clouds coupled to the surface versus those
uncoupled, but did not conduct in-situ observations of the microphysics within the cloud, and





vertical aerosol characterization was constrained to particles >300 nm. Hobbs and Rango (1998)
found that droplets in low clouds over the Beaufort Sea in June occasionally contained drops as
large as 35 μm diameter. They also found that the CDNC in the cloud tops correlated
significantly with "aerosols" below the bases. They suggested that cloud-top entrainment did not
control the CDNC, although there may be times when entrainment does influence the Arctic
CDNC (e.g. Klingebiel et al., 2014).

Motivated by the limited knowledge of aerosol effects on cloud in the summer Arctic and

the details of particle activation, the Canadian Network on Climate and Aerosols: Addressing
Key Uncertainties in Remote Canadian Environments (NETCARE - http://www.netcare-
project.ca/), conducted airborne observations of aerosols and clouds during July, 2014 in the area
around Resolute Bay, Nunavut, Canada. The observations from the study are used here to
characterize CDNC, LWC, and the volume-weighted mean droplet diameter (VMD) relative to
measured aerosol particle size distributions (5 nm and larger) and CCNC. In particular, the
following questions are addressed:
1) Given the scarcity of data, what are the characteristics of clouds in the summertime Arctic,

and do clouds near the surface have characteristics different from those aloft? (Sect. 3.2)

2) What are the sizes of particles that are acting as nuclei for cloud droplets, which will allow a

closer connection between aerosol processes, particle sizes and climate effects? (Sect. 3.3)

3) What is the relationship between droplet size and droplet number? In particular, what is the

aerosol influence on cloud below the Mauritsen-limit, and is it possible to assess a

background influence of the aerosol on clouds in the Arctic summer? (Sect. 3.4)




**2.    Methodologies**

The instrument platform was the Alfred Wegener Institute (AWI) Polar 6 aircraft, a DC-3
aircraft converted to a Basler BT-67 (see Herber, A., Dethloff, K., Haas, C., Steinhage, D.,
Strapp, J. W., Bottenheim, J., McElroy, T. and Yamanouchi, T.; POLAR 5 - a new research
aircraft for improved access to the Arctic, ISAR-1, Drastic Change under the Global Warming,
Extended Abstract, pp. 54-57, 2008).

**2.1    Instrumentation**

The following measurements are relevant to this discussion:
a)  Particle number concentrations >5 nm diameter were measured with a TSI 3787 water-

based ultrafine condensation particle counter (UCPC), sampling at a flow rate of 0.6 L

$min^{-1}$. Hereafter, the measurements are referred to as N5.

b)  Aerosol particle size distributions from 20 nm to 100 nm (45 s up scans and 15 s down

scans)   were measured using a Brechtel Manufacturing Incorporated (BMI) Scanning

Mobility System (SMS) coupled with a TSI 3010 Condensation Particle Counter (CPC).

The sheath and sample flows were set to 6 L $min^{-1}$ and 1 L $min^{-1}$. BMI software was used

to process the distributions.

c)  Aerosol particle size distributions from 70 nm to 1 μm were measured using a Droplet

Measurement Technology (DMT) Ultra High Sensitivity Aerosol Spectrometer (UHSAS)

that uses scattering of 1054 nm laser light to detect particles (e.g. Cai et al., 2008).





159    d) CCNC were measured using a DMT CCN Model 100 counter operating behind a DMT

low pressure inlet at a reduced pressure of approximately 650 hPa. For the nominal water

supersaturation of 1%, the effective supersaturation at 650 hPa was found to be 0.6% as

discussed below. The supersaturation was held constant throughout the study to allow for

more stability of measurement, improved response, and to examine the hygroscopicity of

smaller particles.

e) Droplet size distributions from 2-45 μm were measured using a Particle Measuring

Systems (PMS) FSSP-100. This FSSP-100 had been modified with new tips to reduce

shattering artifacts (Korolev et al., 2011), and it was mounted in a canister under the port-

side wing. The CDNC, VMD and LWC are calculated from the measured droplet

distributions.

f) Cloud particle images in two dimensions for particles sized from about 50 μm to 800 μm

were measured using a PMS 2DC Grey-scale probe. For the present study, these

observations are used only to ensure the absence of the ice phase. This 2DC-Grey was

also modified with new tips to reduce shattering artifacts (Korolev et al., 2011), and it

was mounted in a canister beside the FSSP-100.

175    g) Carbon monoxide (CO) is used here as a relative indicator of aerosol influenced by

pollution sources and as a potential tracer for aerosol particles entering cloud. The CO

was measured with an Aerolaser ultra-fast carbon monoxide monitor model AL 5002

based on VUV-fluorimetry, employing the excitation of CO at 150nm. The instrument

was modified such that in-situ calibrations could be conducted in flight.


Details of the instrument calibration and evaluations are given in the Supplement (S1).




## 2.1   State parameters and Winds


State parameters and meteorological measurements were measured with an AIMMS-20,
manufactured by Aventech Research Inc. The instrument consists of three modules: 1) an Air
Data Probe that measures the three-dimensional aircraft-relative flow vector (true air speed,
angle-of-attack, and sideslip), temperature and relative humidity, and includes a three-axis
accelerometer pack for turbulence measurement; 2) an Inertial Measurement Unit that consists of
three gyros and three accelerometers providing the aircraft angular rate and acceleration; 3) a
Global Positioning System for aircraft 3D position and inertial velocity. Horizontal and vertical
wind speeds are measured with accuracies of 0.50 and 0.75 m/s, respectively; the vertical
resolution was insufficient to measure gusts in the sampled clouds. The accuracy and resolution
for temperature measurement are 0.30 and 0.01 C. The accuracy and resolution for relative
humidity measurement are 2.0 and 0.1 %. The sampling frequency is 1 Hz.

## 2.2   Inlets


Aerosol particles were sampled through a shrouded inlet diffuser (diameter of 0.35 cm at intake
point), which is the same inlet discussed by Leaitch et al. (2010). Transmission of particles by
the inlet is approximately unity for particles from 20 nm to <1 μm and at the airspeeds operated
at here. The intake was connected inside the cabin to a 1.9 cm OD stainless steel manifold off
which sample lines were drawn to the various instrument racks using angled inserts. The total
flow at the intake point was approximately isokinetic at 55 L min$^{-1}$ based on the sum of flows





drawn by the instrumentation (35 L min⁻¹) and the measured flow at the exhaust of the tube. The
flow at the exhaust of the tube was allowed to flow freely into the back of the cabin so that the
flow at the intake varied by the aircraft TAS and the manifold was not significantly over
pressured.

CO was sampled through a separate inlet consisting of a 0.40 cm OD Teflon tube using

the forward motion of the aircraft to push air into the line in combination with a rear-facing 0.95
cm OD Teflon exhaust line that lowered the line pressure. The continuously measured sample
flow was approximately 12 L min⁻¹.

**2.3     Approach to Analysis**

Eleven research flights were conducted from Resolute Bay, Nunavut (74°40'48"N 94°52'12"W)
beginning July 4 and ending July 21, 2014. The measurements were associated with two
relatively distinct weather regimes. During period 1 (July 4-12), the weather conditions around
Resolute Bay were affected by an intensive upper low (Supplement Fig. S4). The wind speeds at
500 hPa were mostly calm and varying from south to north. The surface (1000 hPa) was
dominated by weak high-pressure with generally clear skies, light winds, and occasional
scattered to broken stratocumulus. Low-cloud or fog was at times present in association with
open water, and the air was relatively clean as discussed below. There was a transition period
from July 13-16 when flights were not possible due to fog at Resolute Bay. During period 2 (July
17-21), the area came under the influence of a deep low pressure system to the south
(Supplement Fig. S5) that brought more wind and higher cloud. The air was not as clean as
during period 1, based on the measured aerosol mass and CO concentrations (see Table 1), in



part due to transport of BB aerosol from the Northwest Territories by the low (Köllner et al.,
2015). A separate analysis, based on the bulk Richardson number and data from radiosondes,
estimated boundary-layer heights at 254 m (±155 m) (Aliabadi et al., 2015).

A summary of all flight tracks is shown in Fig. 1. Flights mostly consisted of vertical

profiles and of low level flying over ice, water and melt ponds that contributed to the formation
of low cloud, where low cloud is defined here as cloud tops below 200 m above the surface.
Higher level cloud was also sampled during the profiles and transits. The polynyas that were
sampled over are shown in the top center of each panel of Fig. 2. Cloud was sampled on eight of
the 11 flights, more frequently during period 1 because of better visual contrast at lower levels
and because period 2 marked the presence of the Canadian Coast Guard Ship Amundsen in
Lancaster Sound (bottom center of each panel in Fig. 2) when the flight plans were focused
towards sampling of the ship's plume.

All aerosol number concentrations are given in terms of standard atmospheric pressure

and temperature (STP: 1 atm and 15°C). The CDNC are also referenced to STP where
comparisons are made with the aerosol number concentrations. Number concentrations greater
than 100 nm (N100) are taken from the UHSAS. All data, except for the SMS, are 1 second
averages that represent a sampling path length of 60–80 m. The size distributions over 20-100
nm are from the SMS data, which are 1-minute averages. Except for the example shown in Fig.
S3, all number concentrations of particles smaller than 100 nm are taken from the SMS. Nx-100
refers to the number concentration within the interval "x-100" where x ranges between 20 and
90.  Values of Nx with x < 100 are derived from the sum of Nx-100 (SMS) + N100 (UHSAS).

Clouds were sampled when they were present in the area of study, ideally by ascending

or descending through them.  The latter was usually possible for the higher-altitude clouds but



seldom possible for the low-altitude clouds.  Nearly all clouds sampled were liquid phase, and
only liquid phase clouds are discussed here based on based on the 2DC-Grey images. Droplet
sizes are represented by the volume-weighted mean diameter (VMD), which has the property
that the VMD can be used with the CDNC to calculate the LWC.
The pre-cloud aerosol is determined from averages of values collected within about 50 m
of cloud base where a cloud base was clear and achievable. In other cases the pre-cloud aerosol
are the values at the similar or lower altitudes in the clear air upwind of the cloud.  For the
aerosol measurements made with the 1-minute averaged number concentrations from the SMS,
values from further below-cloud are necessary in some cases. These values are however
consistent with the 1-second aerosol measurements closer to cloud base.  In sections 2.4.1 and
2.4.2, detailed examples are used to show how the aerosol observations relate to the cloud
observations for the higher-altitude (HA) cloud (clouds based above 200 m) and low-altitude
(LA) cloud (clouds topped below 200 m), and to demonstrate how the pre-cloud aerosol
concentrations were assessed.

**2.4.1       Higher Altitude (HA) Cloud Examples**

Four examples of profiles through HA clouds are shown in Fig. 3. There are two panels for each
profile: the left-hand panel shows CO, CDNC and particle number concentrations (N5, Nx-100,
N100, CCNC); the right-hand panel shows temperature, equivalent potential temperature ($\theta_e$),
LWC and VMD. The temperatures, $\theta_e$ and VMD are scaled as indicated.
July 7 Case (Fig. 3 a, b): One of several profiles through a stratocumulus layer on July 7
sampled during the transits to and from the polynyas north of Resolute Bay. The CDNC (at STP)





are relatively constant with altitude and the LWC and VMD both increase steadily with altitude,
features common to the formation of cloud by lifting of air and indicating that the cloud droplets
were nucleated on particles in air rising from cloud base. The cloud top is capped by a
temperature inversion of about $2^{o}C$ at 2350 m, and the particle profiles along with $\theta_e$ and CO are
relatively constant below the cloud base. In-cloud, the number concentrations of larger particles
(N100) is reduced due to scavenging by the cloud droplets; although such particles are not
completely eliminated as smaller droplets can enter the inlet and dry in the sampling lines.
Smaller particles can be artificially increased in cloud due to the shattering of larger droplets on
the aerosol intake, as indicated by the increase in the N5 higher in the cloud. The in-cloud
aerosol measurements are shown here only for completeness, but they are not used in the
subsequent analysis. The CDNC range up to 265 cm$^{-3}$ and the mean value is 199 cm$^{-3}$. Below
cloud base, the N5, N20-100, N30-100, N50-100, N100 and CCNC are approximately 235 cm$^{-3}$,
167 cm$^{-3}$, 145 cm$^{-3}$, 94 cm$^{-3}$, 67 cm$^{-3}$ and 117 cm$^{-3}$, respectively. The below-cloud N20 of 234
cm$^{-3}$ approximately equals the N5 offering confidence in terms of the closure of number
concentrations. The N30 values compares most closely with the mean CDNC leading to the
conclusion that in the mean cloud droplets nucleated on particles down to about 30 nm; see
Supplement Fig. S6a. It is possible that particles as small as 20 nm contributed to the CDNC in
this cloud based on the maximum CDNC; for 20 nm particles to activate, the water
supersaturations in the bases of the clouds would have had to reach about 1.5%. The below-cloud
number size distribution for this case is compared with a below-cloud case from over the western
Atlantic Ocean (S.I. Fig. S6a). The overall particle concentration is much higher in the Atlantic
Ocean case, and the activation size is correspondingly higher at about 110 nm. There are some
differences in activation due to differences in updraft speeds, but the volume concentration of





particles larger than 100 nm in the Atlantic case is more than 30 times higher than the present
case of July 7 (S.I. Fig. 6b). The much lower volume concentration of larger particles in the July
7 case will lower water uptake during the initial stages of growth, resulting in a higher maximum
supersaturation.
July 17 Case (Fig. 3 c, d): The July 17 profile is similar in some respects to the July 7
case, but there is more variability near the cloud top and the CDNC (STP) are overall  lower with
a maximum of about 75 cm$^{-3}$ and a mean of 55 cm$^{-3}$. The VMD exceed 20 μm compared with a
peak value of about 15 μm for the July 7 profile. Continuity from about 100 m below cloud base
is indicated by the CO and $\theta_e$ profiles, and the N50 approximates the mean CDNC and possibly
the maximum CDNC. The CCNC are 30-40 cm$^{-3}$ below cloud, indicating a supersaturation larger
than 0.6%.
July 19 Case (Fig. 3 e, f): The July 19 profile includes two cloud layers, one from 1200-
1400 m and the second from 1400-1500 m. The layer separation appears in the CO
concentrations, which are uniform through the lower layer and increasing in the upper layer. The
CO levels of 100+ ppbv in this case are among the highest observed during the study, reflecting
contributions from BB as discussed by Köllner et al. (2015).  The mean CDNC (STP) in the
lower and upper layers are 239 cm$^{-3}$ and 276 cm$^{-3}$ respectively.  The VMD reach 15 μm in the
lower layer and are smaller in the upper layer. The smaller VMD is influenced by the lower
LWC and the slightly higher CDNC, where the latter may be governed by the increase in aerosol
from below the layer to above the layer.  The N50 and N100 estimated for the lower (upper)
layer are 269 (334) cm$^{-3}$ and 197 (221) cm$^{-3}$ respectively. Thus, in the mean the CDNC in both
layers are approximated by the activation of particles sized between 50 nm and 100 nm, and the
maximum CDNC suggests activation of particles down to about 50 nm. The CCNC are slightly



below the N100 possibly due to a reduced hygroscopicity of the BB particles. Comparison of the
CCNC with the CDNC suggests cloud supersaturations above 0.6%.
July 20 Case (Fig. 3 g and h): A more complex cloud with substantial variations in the
LWC that suggests three cloud layers. The values of the mean CDNC at STP are 45 cm$^{-3}$, 49 cm$^{-3}$
and 65 cm$^{-3}$ in the upper, middle and lower layers respectively.  The VMD reach about 20 μm
in the lower layer and 26 μm in the upper layer with the lower CDNC. The layers are relatively
stable with CO and θ$_e$ increasing slightly from below the cloud to above the top cloud layer.  The
N50 just below the lower layer approximately equal the CDNC in that layer. It is more difficult
to estimate the pre-cloud aerosol for the middle and upper layers, but particles at least as small as
50 nm were activated.  For the summary statistics, the respective pre-cloud N100, N50 and
CCNC are estimated at 24 cm$^{-3}$, 44 cm$^{-3}$ and 24 cm$^{-3}$ for the upper cloud layer, 32 cm$^{-3}$, 52 cm$^{-3}$
and 32 cm$^{-3}$ for the middle layer and 34 cm$^{-3}$, and 66 cm$^{-3}$ and 35 cm$^{-3}$ for the lower layer.
Comparison of the CCNC with the CDNC suggests supersaturations near or in excess of 0.6%.

**2.4.2         Low-Altitude (LA) Examples**

July 5 and July 7 Cases: The two examples in Fig. 4 are for cloud or fog over the polynyas north
of Resolute Bay on July 5 and July 7. Four cloud samples were collected on July 5 at altitudes
below 200 m.  The time series in Fig. 4a covers the period of collection of the two lowest
samples: 16:18:02-16:21:57 at 130 m and 16:39:35-16:40:18 at 88 m.  In the air upwind of the
cloud or fog, the N100, N30 and CCNC are estimated at 3 cm$^{-3}$, 10-14 cm$^{-3}$ and 5 cm$^{-3}$. The
mean values of the CDNC of 2.8 cm$^{-3}$ at 130 m and 0.7 cm$^{-3}$ at 88 m are explained by the N100
and supersaturations less than 0.6%. The maximum CDNC of 12 cm$^{-3}$ at 130 m suggests the



activation of smaller particles, possibly as small as 30 nm, and a supersaturation exceeding 0.6%
perhaps due to some uplift influenced by orographic features north of the north polynya. At 88
m, the mean VMD (not shown) was 29 μm and ranged up to 35 μm giving those droplets
potential to deposit over an hour or more, thereby transferring water from the polynya to the
downwind ice. On July 7, cloud or fog was present below 120 m and thicker towards the north
edge of the north polynya and again to the north over the ice. The CDNC are overall higher with
averages of seven samples over the period 16:06-16:29 ranging from 4 cm$^{-3}$ to 13 cm$^{-3}$, the one-
second CDNC are as high as 34 cm$^{-3}$ and the mean VMD (not shown) range from 19.6 μm to
22.8 μm. The CO mixing ratio is slightly higher in the cloud (81 ppbv) than above (79 ppbv).  In
the air nearly free of cloud and below 120 m, the N100 are 4-5 cm$^{-3}$, the N50 are 8-11 cm$^{-3}$ and
the N20 are variable between 17 cm$^{-3}$ and 130 cm$^{-3}$; CCN are unavailable for this part of the
flight due to instrument problems.  Mean values of CDNC/N100 and CDNC/N50 for seven cloud
samples are 4.8 and 1.0, respectively, indicating that on average particles about 50 nm were
activated in this LA cloud. Based on the overall relationship between CCNC and N50, which is
discussed in section 3.3, the mean supersaturation in the LA cloud of July 7 is estimated at 0.6%.
Comparison with the maximum CDNC suggests that particles as small as 20 nm may have
participated in the nucleation of droplets.

July 8 Case:  Fig. 5 shows a time series of altitude, CO, N100, N80-100, N90-100, CCNC

and CDNC from the sampling above and in the low cloud over Lancaster Sound on July 8. The
cloud over the open water of the Sound is visible in the satellite picture in Fig. 2b. Cloud was
also present over the ice to the west, but it was much thinner and reached only to about 150 m.
Over the water, the cloud was sampled as high as 230 m by descending into it down to about 150
m between 17:27 UT and 17:43 UT as shown in Fig. 5. Observations in profiles from two of the



five samples are shown in Fig. 6. The cloud deepened as the aircraft approached the ice edge
from over the water, and thinned abruptly over the ice with tops below 150 m as shown in Fig. 5
(time 17:47). The thicker cloud was associated with a shift in the wind direction to be more
southerly suggesting an influence of the Prince Regent Inlet and surrounding terrain on the flow
as well as possibly circulations influenced by the water-ice transition. The cloud layer was
relatively stable and the $\theta_e$ profiles suggest a surface heat sink (Fig. 6a). The profiles of LWC
and VMD in Fig. 6 (b, c) do not show increases with altitude characteristic of vertical mixing,
such as for some of the HA clouds (Fig. 3); the change in the VMD per 50 m increase in height is
about 1.7 μm for the well mixed cloud of July 7 (Fig. 3 a, b), whereas it is about 0.2 μm per 50 m
for the LA cloud of flight 8 in Fig. 6.  The CO mixing ratio shows little variation with time and
altitude. The pre-cloud aerosol concentrations are more difficult to assess.  Based on
concentrations just above the cloud, particles >90 nm explain the CDNC. Based on the
concentrations downwind at 150 m (approximately 17:47), activation of particles >80 nm is
needed to explain the CDNC. The CCNC are about 129 cm$^{-3}$ downwind and between 157 cm$^{-3}$
and 234 cm$^{-3}$ just above cloud. It is concluded that in this case the droplets likely nucleated on
particles mostly larger than 80-95 nm and the supersaturations in the clouds were less than 0.6%.
For the purposes of summary statistics discussed next, the N100, N50 and CCNC have been
selected as an average of the downwind and immediately above cloud concentrations: 73 cm$^{-3}$,
319 cm$^{-3}$ and 168 cm$^{-3}$, respectively.






**3.    Summary Observations and Discussion**

Summary statistics for the cloud and aerosol samples are discussed in 3.1, the microphysics of
low-altitude and higher-altitude clouds are contrasted in 3.2, particle activation is summarized in
3.3 and in section 3.4 the relationship between VMD and CDNC is used to consider the
transition of aerosol indirect effects from potential warming to potential cooling.

**3.1    Summary of mean observations**

A total of 62 liquid water cloud samples were averaged where the mean LWC was $> 0.01$ g m$^{-3}$.
The samples are integrations over periods ranging from 11 seconds to 1000 seconds with a
median sample time of 65 seconds that is equivalent to a horizontal path length of about four
kilometers. The mean and median values of the microphysical properties of the aerosols and
clouds as well as the altitudes and temperatures associated with the 62 cloud samples are given in
Table 1, separated between periods 1 and 2. Values of the CDNC and the LWC are given relative
to in-situ volumes as well as STP. As discussed above, the pre-cloud CCNC, N50, and N100 are
averages of those values collected within about 50 m of cloud base where a cloud base was clear
and achievable. In other cases the pre-cloud CCNC, N50 and N100 are the values at the similar
or lower altitudes in the clear air upwind of the cloud, except in the case of July 8 when the pre-
cloud aerosol is based on the measurements in the area downwind plus those immediately above
cloud. The CCNC samples in Table 1 are limited to 44 due to instrument problems, all of which
occurred during the early part of July 7.



Cloud liquid water paths (LWP) are estimated for 36 of the samples when a complete
profile between cloud base and cloud top was possible. The LWP are shown at the bottom of
Table 1. Thirty-four of the 36 LWP estimates are for above 200 m, and the mean and median
altitudes are 1044 m and 862 m, respectively. Not included in the summary statistics are the
samples from July 8 shown in Fig.s 5 and 6. For the minimum altitude reached in that cloud, the
LWP ranged from 12 to 25 and thus the total LWP for that cloud ranged to above 25.
During period 1, the median sampling altitude is lower and the temperatures are slightly
below freezing compared with just above freezing during period 2. The CO mixing ratios are
overall low and at approximately background values during period 1. The median CDNC are
higher during period 1 than period 2, but the mean values are similar.  The CDNC compare more
closely with the N50 during period 1, while during period 2 the CDNC are about equally
between the N50 and N100. The CCNC equated with particles between 50 nm and 100 nm
during period 1, whereas during period 2 they were closer to the N100 values. The reduction in
particle hygroscopicity during period 2 may be due to an increased presence of organics in the
aerosol during that time (Willis et al., personal communication).

**3.2    Comparison of LA and HA cloud**

The LA clouds are associated with low-level horizontal advection and heat and water exchange
with the underlying ice or water surface. In contrast, vertical motions are responsible for some of
the HA clouds, and none of the HA clouds interact so closely with the underlying surface. Due to
those differences, the characteristics of the LA and HA clouds are considered separately. Table 2
shows the mean and median values for the samples separated between LA and HA clouds;



vertical profiles of the CDNC, LWC and VMD samples are shown in Supplement Fig. S7.  On
average, the LA samples have lower CDNC and higher VMD compared with the HA cases, and
the LA clouds are activating on larger particles relative to the HA clouds (e.g. CDNC/N50). The
values of the CDNC/CCNC indicate that the supersaturations are <0.6% for the LA clouds and
close to 0.6% for the HA clouds.

Variations in LWC are correlated with those of CDNC for the LA samples (Fig. 7a). The

coefficient of determination ($R^2$) rises from 0.57 to 0.98 if the one LA point at (137, 0.032) is
removed. In contrast, the correlation of the LWC with the CDNC for the HA samples is low
($R^2$=0.16). There is no correlation of the LWC with the VMD for the LA points ($R^2$=0.04), and
for the HA clouds there is a modest correlation of LWC with MVD ($R^2$=0.26). Variations in
LWC with VMD within a cloud system are consistent with lifting of air from below, i.e.
nucleation of droplets at cloud base followed by their growth with increasing altitude, such as the
case shown in Fig. 3a and 3b. They can also result from homogeneous mixing (i.e. entrainment
of dry air that reduces the LWC by evaporation of the droplets without reducing the CDNC). The
strong dependence of the variations in LWC with those of the CDNC in the LA clouds may
reflect changes in rate of cooling, collision-coalescence or inhomogeneous mixing along the
cloud transport pathway. For example, increases in the rate of cooling within or between clouds
will increase condensation rates, and potentially supersaturations, resulting in increased LWC
and CDNC. Reductions in the collision-coalescence process will result in higher CDNC and
lower deposition rates that can increase the LWC.  Inhomogeneous mixing, the entrainment of
dry air parcels into a cloud without mixing with the cloud droplets, will reduce the CDNC
averaged across the cloud and at the same time reduce the mean LWC. Changes in the aerosol,




which are interactive with some of the cloud processes, may contribute to the CDNC and
potentially the LWC through their influence on collision-coalescence.

The LWC-CDNC correlation is identifiable for individual flights with sufficient LA

samples: four flights, comprising 20 of the 24 LA samples, had three or more points as shown in
Fig. 8. The regressions for each of the July 7, 8 and 17 cases are approximately linear, and the
respective mean VMD are 20.8 μm, 18.8 μm and 18.2 μm. The mean LWC are 0.05 g m$^{-3}$, 0.3 g
m$^{-3}$ and 0.07 g m$^{-3}$. The VMD are relatively close together confirming similarities in the
relationship, even if it not purely linear.  For comparison, the mean VMD for the July 5 samples
is 29.2 um and the LWC is 0.02 g m$^{-3}$, which indicates that the July 5 case does not fit the linear
relationship shown in Fig. 8a. The reasons behind the similarity of the VMD for the July 7, 8 and
17 are unknown, but it occurs despite the varied pre-cloud N50 and N100: N50 range of 5-272
cm$^{-3}$; N100 range of 1.1-73 cm$^{-3}$. The consistencies among the three flights for greatly differing
aerosol and CDNC imply a much smaller role for the aerosol in terms of the LWC. The
distributions of droplets extend above 20 μm in these cases, but few are of sufficient size to
initiate collision-coalescence (about 30 μm) (e.g. Rosenfeld et al., 2001) unless some fall out
already had occurred.  Greater temporal and spatial coverage are needed to assess the
microphysical processes in these clouds.

**3.3    Particle Activation Sizes**

Here, the sizes and CCN activity of particles that acted as nuclei for cloud droplets are examined.
As shown in the plot of CDNC versus N100 (Fig. 9a), particles smaller than 100 nm activated in
most cases and most often in the HA clouds. The mean and median values of CDNC(STP)/N100



are 2.2 and 1.8 for all 62 samples, and the 30[th] percentile of the CDNC/N100 is 1.2, which means
that in about 70 % of the cases droplets nucleated on particles significantly smaller than 100 nm.
Fig. 9a can be compared with the results of Hegg et al. (2012) who showed a linear fit of CDNC
to N100 for marine stratocumulus with a slope of 0.72 for which the N100 in 94% of the samples
was >150 cm$^{-3}$. Here, a slope larger than unity is indicated, and the N100 are <100 cm$^{-3}$ in 90%
of the samples. The comparison indicates that relationships derived for higher concentration
environments do not necessarily apply to those to lower concentration environments. In the clean
environment often found in the Arctic during summer, the absence of larger particles may lower
water uptake rates during droplet nucleation, which will increase the supersaturation, enabling
cloud droplets to nucleate on smaller particles; the absence of larger particles may also help
increase the concentrations of smaller particles in the Arctic during summer, by promoting new
particle formation through a reduced condensation sink (e.g. Leaitch et al., 2013). The CDNC are
plotted against the N50 in Fig. 9b showing that the mean activation size was often close to 50
nm.  The median value of CDNC/N50 is 0.78 for all samples indicating that, based on the
averaged CDNC, cloud droplets nucleated on particles near or smaller than 50 nm about 40% of
the time.  That percentage will increase if particle activation is considered relative to the
maximum CDNC.

The mean and median values of the CCNC associated with all cloud samples (84 cm$^{-3}$

and 47 cm$^{-3}$) are generally consistent with previous Arctic CCNC measurements. For example,
during the summer above 85$^{o}$N, Martin et al. (2011) measured a mean CCNC at 0.73%
supersaturation of 47 cm$^{-3}$ with a standard deviation of 35 cm$^{-3}$, and Yum and Hudson (2001)
measured CCNC at 0.8% supersaturation below 1700 m over the Beaufort Sea during May, 1998
that ranged from 41 cm$^{-3}$ to 290 cm$^{-3}$. Considering the median values of CDNC/CCNC for the



LA and HA samples (Table 2) and the slopes of linear regressions of CDNC versus CCNC (Fig.
10a), the average inferred supersaturation for the HA clouds is about 0.6%, consistent with the
overall activation of smaller particles in those clouds. The mean supersaturation inferred for the
LA clouds is significantly lower than 0.6%. Based on the activation of a 90 nm particle (July 8
case; CCNC of 168 cm$^{-3}$ in Fig. 10a) of low-moderate hygroscopicity, a reasonable estimate is
0.3% for the mean of the LA clouds with some higher values indicated by the points near a
CCNC of 25 cm$^{-3}$ in Fig. 10a. The supersaturations for these clean clouds are in contrast to
polluted marine environments for which estimates for these types of clouds are 0.2% or less (e.g.
Modini et al., 2015). Consistent with the present results, Hudson et al. (2010) found that effective
supersaturations in marine stratus tended to increase with a decrease in the CCNC, and for
CCNC smaller than about 200 cm$^{-3}$ their effective supersaturations ranged between 0.3% and

1.2%.

Variations in the measured CCNC are explained well by variations in smaller (N50) and

larger (N100) particles as shown in Fig. 10b. The slopes of the power-law fits, for which the
exponents are both close to unity, indicate that the CCNC at 0.6% supersaturation on average fall
between 50 nm and 100nm.

**3.4      Aerosol Influences on Warming to Cooling**

Here, the aerosol influence on clouds with CDNC below the Mauritsen-limit and the potential
background influence of the aerosol on clouds with CDNC above the Mauritsen-limit are
considered. The relationship between the VMD and CDNC, shown in Fig. 11, exhibits a
scattered but clear tendency for smaller VMD with increasing CDNC.  The solid black curve is a





reference line based on the study-mean LWC of 0.12 g m$^{-3}$ (Table 1); points falling above or
below the black curve have higher or lower LWC, respectively. The vertical dashed green line
represents the Mauritsen limit below which the cloud may produce a net warming for an increase
in the CDNC. The net warming is a consequence of an increase in longwave absorption due to an
increase in the LWC, where the latter results from a reduction in deposition for the smaller
droplets associated with increased CDNC. Above the Mauritsen limit, an increase in CDNC may
produce a net cooling due to the cloud albedo effect, since at that point the longwave forcing
does not change significantly as the effects of deposition are minimized and the cloud effectively
behaves as a black body.

**3.4.1        Below the Mauritsen limit**

Seventeen of the 62 samples fall at or below the Mauritsen limit, which is suggested here as 16
cm$^{-3}$. The value of 16 cm$^{-3}$ is selected because all points fall below the mean LWC, and these
points offer greater potential for changes in the CDNC to increase the LWC. Fifteen of the 17
samples are from LA clouds with median pre-cloud N50 and N100 estimates of 8.2 cm$^{-3}$ and 3.0
cm$^{-3}$ respectively.  The lower number concentrations contribute to overall larger VMD; although
some of the points below the Mauritsen limit have VMD values much less than 20 μm. Increases
in small particles, potentially from particle nucleation or fragmentation (e.g. Leck and Bigg,
1999 and 2010), are hypothesized to increase the CDNC thereby enhancing the longwave
warming by these clouds at least until the CDNC reach above the Mauritsen limit. The LA points
from the July 5 and the July 7 cases, identified in Fig. 11, offer one insight. Considering the
median CDNC for July 5 is eight times lower than that for July 7, the absence of a similar



difference in the N50 and N100 for July 5 and July 7 (see Fig. 11 caption) suggests the aerosol
was not a factor in the differences in CDNC. Consistent with the discussion in section 3.2, all 15
LA points show a correlation of LWC with the CDNC ($R^2$=0.57), but correlations of the CDNC
with the N50 and N100 are weak: $R^2$=0.19 and 0.06, respectively.  Because only seven points
with CCNC are available, which correlate well with the N50, the CCN are not used here. If the
Mauritsen limit of 10 $cm^{-3}$ is applied, reducing the number of points to 12, the assessment does
not change: the LWC-CDNC correlation improves slightly and the correlations of the CDNC
with the N100 and the N50 weaken. Also, the LWC do not correlate with either the N50 or the
N100 (Supplement Fig. 8). In this low CDNC environment, where cloud droplets may grow
large enough to be gravitationally removed from the cloud without the support of collision-
coalescence, the absence of a positive correlation of either the CDNC or LWC with the aerosol
indicates that changes in the aerosol did not contribute significantly to the changes in the LWC.
Variations in other processes, such as mixing or the rate of cooling, may be responsible for the
correlation of CDNC and LWC. In these cases, there is sufficient aerosol so as not to be a
limiting factor for the CDNC.

**3.4.2          Background aerosol influence on clouds**

Above the Mauritsen limit, the general reduction in the VMD with the CCNC-associated
increase in CDNC reflects the aerosol impact on the clouds. In Fig. 11, samples are identified
between those associated with lower CO (green circles; <81 ppbv, the median CO value of all
samples) and those with highest CO (red circles; >90 ppbv); six samples have no CO
measurement and the remaining points have CO falling within 81-90 ppbv. Five of the seven



higher-CO samples are from the July 19 case (e.g. Fig. 3e, 3f) that has been linked with BB
(Köllner et al., 2015; reference above), and the highest CDNC point (273 cm$^{-3}$; no CO
measurement) is also from July 19 and likely influenced by BB. The higher-CO samples cover a
range of CDNC from 16 cm$^{-3}$ to at least 238 cm$^{-3}$ with CO reaching up to 113 ppbv. The higher
CO samples are associated with larger particles (N50/N100=1.5), consistent with a BB influence,
compared with the lower CO samples (N50/N100=3.2). These values for BB fall at the low end
of the observations from Zamora et al. (2015), but their CO concentrations are much higher than
those measured in this study. The lower-CO samples may be dominated by regional biogenic
emissions (Burkart et al., 2015). The lower- and higher-CO points overlap over a CDNC range
of 16 cm$^{-3}$ to 160 cm$^{-3}$, consistent with the range of pre-industrial CDNC from global models of
30 cm$^{-3}$ to 140 cm$^{-3}$ (Penner et al., 2006) and suggesting that 20-100 nm particles from natural
sources can have a broad impact on the range of CDNC in clean environments. This result
affirms the large uncertainty associated with estimating a baseline for the cloud albedo effect
discussed by Carslaw et al. (2013).

**4.    Summary and Conclusions**

Aerosol particle size distributions, CCNC at 0.6% water supersaturation, carbon monoxide (CO)
and cloud microphysics were measured from an airborne platform based out of Resolute Bay,
Nunavut from July 4 to July, 21, 2014 as one part of the Canadian NETCARE project. The
flights were conducted over ice and water surfaces from about 60 m above the surface to about
6000 m. Sixty-two (62) cloud-averaged samples were derived, each constrained for the mean
LWC >0.01 g m$^{-3}$. The analysis separates the cloud samples between 24 low-altitude (LA: <200



m) samples and 38 higher altitude (HA: >200 m) samples as well as situations of lower and
higher CO and observations above and below the Mauritsen et al., (2011) CCNC (or CDNC)
limit.

The median pre-cloud N100 of 33 cm$^{-3}$ and the median CO mixing ratio of 81 ppbv

indicate that the aerosols supporting the sampled clouds were relatively clean, and particularly
during the first part of the study many of the aerosol particles may have been derived from
regional natural sources. The median CDNC at STP is 10 cm$^{-3}$ for the LA clouds (24 samples)
and 101 cm$^{-3}$ for the HA clouds (38 samples), which correspond with the median pre-cloud N50
of 11 cm$^{-3}$ for the LA samples and 133 cm$^{-3}$ for the HA samples. The lower sizes of particles
activated in cloud varied from about 20 nm to above 100 nm. In 40% of cases, the average lower
size of activation was 50 nm or smaller. Overall, smaller particles were activated more often in
the HA clouds.

From the median values of CDNC/CCNC (1.2 for the HA clouds and 0.6 for the LA

clouds) and linear regression of CDNC and CCNC, it is inferred that the average
supersaturations were approximately 0.6% for the HA clouds and 0.3% for the LA clouds.
Higher estimates will be obtained if the maximum CDNC are taken into consideration rather than
the mean CDNC. The relatively high supersaturations for these clean Arctic stratus and
stratocumulus have similarities with the observations of Hudson et al. (2010) for stratus off the
coast of California.

In 17 cases, 15 of which are LA clouds, the CDNC fell at or below the CCN limit

discussed by Mauritsen et al. (2011), which is estimated here as 16 cm$^{-3}$. These are the first
collection of simultaneous observations of the microphysics of aerosols and clouds in this unique
regime in which the net radiative impact of increases in the CDNC is hypothesized to be



warming due to changes in the LWC. The LWC of the points below the Mauritsen limit all fall
below the study-mean LWC, and the LWC increases with the CDNC. Neither the CDNC nor the
LWC are positively correlated with the pre-cloud aerosol (N50 or N100). In this environment of
low cloud or fog and ultra-low CDNC, variations in cloud processes such as mixing or the rate of
cooling may be responsible for the correlation of CDNC and LWC, whereas the aerosol is
generally sufficient so as not to be the factor limiting the CDNC or LWC.

Forty-five cloud samples with CDNC above the Mauritsen limit exhibit a clear influence

of the aerosol. The cloud microphysics for the clouds formed in cleaner air (smaller particles and
lower CO: <81 ppbv) overlap with clouds formed in more polluted air (larger particles and
higher CO: >90 ppbv) covering a CDNC range of 16-160 cm$^{-3}$. It is concluded that 20-100 nm
particles from natural sources can have a broad impact on the range of CDNC in clean
environments, affirming a large uncertainty in estimating a baseline for the cloud albedo effect.




*Acknowledgements.* The complete data set is available from the NETCARE web site by
contacting Richard Leaitch (Richard.Leaitch@ec.gc.ca) or Jon Abbatt
(jabbatt@chem.utoronto.ca). A spreadsheet containing the details of the 62 samples discussed
here is included with the supplement. The authors acknowledge a large number of people for
their contributions to this work. We thank Kenn Borek Air, in particular Kevin Elke and John
Bayes for their skillful piloting that facilitated these cloud observations. We are grateful to John
Ford, David Heath and the U of Toronto machine shop, Jim Hodgson and Lake Central Air
Services in Muskoka, Jim Watson (Scale Modelbuilders, Inc.), Julia Binder and Martin
Gerhmann (AWI), Mike Harwood and Andrew Elford (EC), for their support of the integration
of the instrumentation and aircraft. We thank Mohammed Wasey for his support of the
instrumentation during the integration and in the field.  We are grateful to Carrie Taylor (EC),
Bob Christensen (U of T), Kevin Riehl (Kenn Borek Air), Lukas Kandora, Manuel Sellmann and
Jens Herrmann (AWI), Desiree Toom, Sangeeta Sharma, Dan Veber, Andrew Platt, Anne Marie
Macdonald, Ralf Staebler and Maurice Watt (EC), Kathy Law and Jennie Thomas (LATMOS)
for their support of the study. We thank the Biogeochemistry department of MPIC for providing
the CO instrument and Dieter Scharffe for his support during the preparation phase of the
campaign. We thank the Nunavut Research Institute and the Nunavut Impact Review Board for
licensing the study. Logistical support in Resolute Bay was provided by the Polar Continental
Shelf Project (PCSP) of Natural Resources Canada under PCSP Field Project #218-14, and we
are particularly grateful to Tim McCagherty and Jodi MacGregor of the PCSP. Funding for this
work was provided by the Natural Sciences and Engineering Research Council of Canada
through the NETCARE project of the Climate Change and Atmospheric Research Program, the
Alfred Wegener Institute and Environment Canada.





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



**Table 1.** Summary of averaged cloud observations with LWC>0.01 g m$^{-3}$ for study periods 1 and
910  2.

| Measurement | Period 1 (July 5-11): 35 samples; 1.2 hours in cloud | | Period 2 (July 11-21): 27 samples; 0.4 hours in cloud | |
|---|---|---|---|---|
| | Mean | Median | Mean | Median |
| Altitude (m-msl) | 920 | 178 | 1011 | 835 |
| Temperature ($^{o}$C) | -1.9 | -0.4 | +1.2 | +2.2 |
| CDNC (STP) (cm$^{-3}$) | 75 (85) | 93 (91) | 73 (83) | 52 (55) |
| LWC (STP) (g m$^{-3}$) | 0.12 (0.13) | 0.10 (0.12) | 0.12 (0.13) | 0.12 (0.13) |
| VMD (μm) | 17.2 | 18.7 | 15.0 | 14.5 |
| CCNC (cm$^{-3}$): 17 P-1; 27 P-2 | 90 | 120 | 81 | 43 |
| N50 (cm$^{-3}$) | 113 | 134 | 126 | 68 |
| N100 (cm$^{-3}$) | 35 | 47 | 81 | 31 |
| CDNC(STP)/CCNC | 0.75 | 0.56 | 1.18 | 1.22 |
| CDNC(STP}/N50 | 0.82 | 0.90 | 0.73 | 0.68 |
| CDNC(STP)/N100 | 2.78 | 2.63 | 1.37 | 1.25 |
| CCNC/N50 | 0.64 | 0.63 | 0.64 | 0.64 |
| CCNC/N100 | 1.92 | 1.79 | 1.27 | 1.0 |
| CO (ppbv) | 79 | 80 | 90 | 87 |
| LWP (g m$^{-2}$); 13 P-1; 23 P-2 | 30 | 27 | 22 | 13 |





**Table 2**. Summary of averaged observations got low-altitude (LA) and higher-altitude (HA) clouds.

| Measurement | LA (<200m): 24 samples; 0.89 hours in cloud | | HA (>200m): 38 samples; 0.72 hours in cloud | |
|---|---|---|---|---|
| | Mean | Median | Mean | Median |
| Altitude (m-msl) | 129 | 127 | 1485 | 1481 |
| Temperature ($^{o}$C) | +0.6 | +0.2 | -1.2 | +0.9 |
| CDNC (STP) (cm$^{-3}$) | 31 (30) | 11 (10) | 101 (118) | 91 (101) |
| LWC (STP) (g m$^{-3}$) | 0.10 (0.10) | 0.05 (0.05) | 0.13 (0.15) | 0.13 (0.15) |
| VMD ($\mu$m) | 20.7 | 20.1 | 13.4 | 12.5 |
| CCNC (cm$^{-3}$); 16 LA; 28 HA | 74 | 24 | 90 | 58 |
| N50 (cm$^{-3}$) | 91 | 11 | 136 | 133 |
| N100 (cm$^{-3}$) | 26 | 4 | 73 | 47 |
| CDNC(STP)/CCNC | 0.61 | 0.57 | 1.2 | 1.2 |
| CDNC(STP}/N50 | 0.61 | 0.44 | 0.91 | 0.93 |
| CDNC(STP)/N100 | 2.3 | 1.4 | 2.1 | 1.9 |
| CCNC/N50 | 0.66 | 0.71 | 0.68 | 0.64 |
| CCNC/N100 | 1.8 | 1.6 | 1.5 | 1.1 |
| CO (ppbv) | 81 | 80 | 86 | 83 |





**Figure. Captions**

Figure 1. Compilation of the flight tracks. All flights originated from Resolute Bay (74°40'48"N 94°52'12"W).

Figure 2. Satellite images from July 5 when LA clouds were sampled over the two polynyas to the north and from July 8 when LA clouds were sampled along Lancaster Sound (July 8). Lancaster Sound is cloud free on July 5 and mostly covered by cloud on July 8. Resolute Bay is marked with a "X". Images are courtesy of NASA Worldview: https://earthdata.nasa.gov/labs/worldview/.

Figure 3. Four examples of profiles through HA clouds. a) Case from July 7 showing CO, CDNC, CCNC and particle number concentrations, where Nx-100, N100 and N5 are for particles sized between "x" nm and 100 nm, >100 nm and >5 nm respectively. b) Case from July 7 showing LWC, VMD, $\theta_e$ and temperature, where VMD, $\theta_e$ and temperature have been scaled as indicated in the legend. c) As in a), but case from July 17 and without N5. d) As in b), but case from July 17. e) As in a), but case from July 19. f) As in b) but case from July 19. g) As in a) but case from July 20 and without N5. H) as in b), but case from July 20. The CDNC are all referenced to STP, and $\theta_e$ is given in degrees Centigrade before scaling.

Figure 4. Time series during the sampling of low (LA) cloud or fog over the polynyas north of Resolute Bay. a) July 5 time series showing CO, CDNC, CCNC and particle number concentrations, where N30-100 is for particles sized between 30 nm and 100 nm and N100 is for particles sized >100 nm. b) July 7 time series showing CO, CDNC and particle number concentrations, where N20-100, N50-100 and N100 are for particles sized between 20 nm and 100 nm, between 50 nm and 100 nm and >100 nm respectively. CCNC measurements are unavailable for this period on July 7. Wind direction and relative position of polynyas are indicated in both panels. CDNC are referenced to STP.

Figure 5. Time series of altitude, CO, N80-100, N90-100, N100, CCNC and CDNC from low cloud (LA) cloud sampling over Lancaster Sound on July 8. The cloud was deeper over the open water of the Sound (see satellite picture in Fig. 2b). Over the ice to the west, the cloud was not as deep and could not be sampled. Segments over water and ice are indicated at the top of the figure.

Figure 6. Profiles down into cloud showing a) $\theta_e$, b) LWC and c) VMDData for periods 17:27-17:29 UT and 17:38-17:39 UT during July 8. d) shows CDNC, N100, CO and CCNC for the 17:27-17:29 UT profile, and e) shows CDNC, N100, CO and CCNC for the 17:38-17:39 UT profile.

Figure 7. The LWC plotted as a function of the CDNC (a) and VMD (b) for the LA (orange) and HA (blue) samples. Linear regressions for each of the LA and HA samples are also plotted, and the coefficients of determination are given in the legends.





Figure 8. As in Fig. 7a, but only for four identified LA cases (July 5, 7, 8 and 11). Linear regressions for each set of samples are also plotted, and the coefficients of determination are given in the legends.

Figure 9. Plots of CDNC versus a) N100 and b) N50. Points are identified between LA (yellow) and HA (blue) samples, and the 1:1 lines are for reference.

Figure 10. a) CDNC plotted versus the CCNC measured at 0.6% supersaturation; points are identified between LA (yellow) and HA (blue) samples, and linear regressions through the origin are shown. b) CCNC plotted versus N50 and N100; power law fits to each are provided for reference.

Figure 11. The mean VMD of all cloud samples plotted versus the CDNC. All CDNC are referenced to the in-situ pressure. The dashed vertical green line represents the "CCN-limited" division discussed by Mauritsen et al (2011) and estimated here as 16 cm$^{-3}$. The solid black line is another reference showing the relationship between VMD and CDNC for a constant LWC: the study mean LWC of 0.12 g m$^{-3}$ (Table 1). Samples with higher CO (>90 ppbv) are identified by the open red circles. Also highlighted for the discussion are LA samples from July 5 (red dots) and July 7 (orange dots). The median CDNC are 1.3 cm$^{-3}$ and 7.8 cm$^{-3}$, for July 5 and 7, respectively; the N50 are 6 cm$^{-3}$ and 8.3 cm$^{-3}$ for July 5 and 7, respectively; the N100 are 3 cm$^{-3}$ and 2.2 cm$^{-3}$ for July 5 and July 7, respectively.



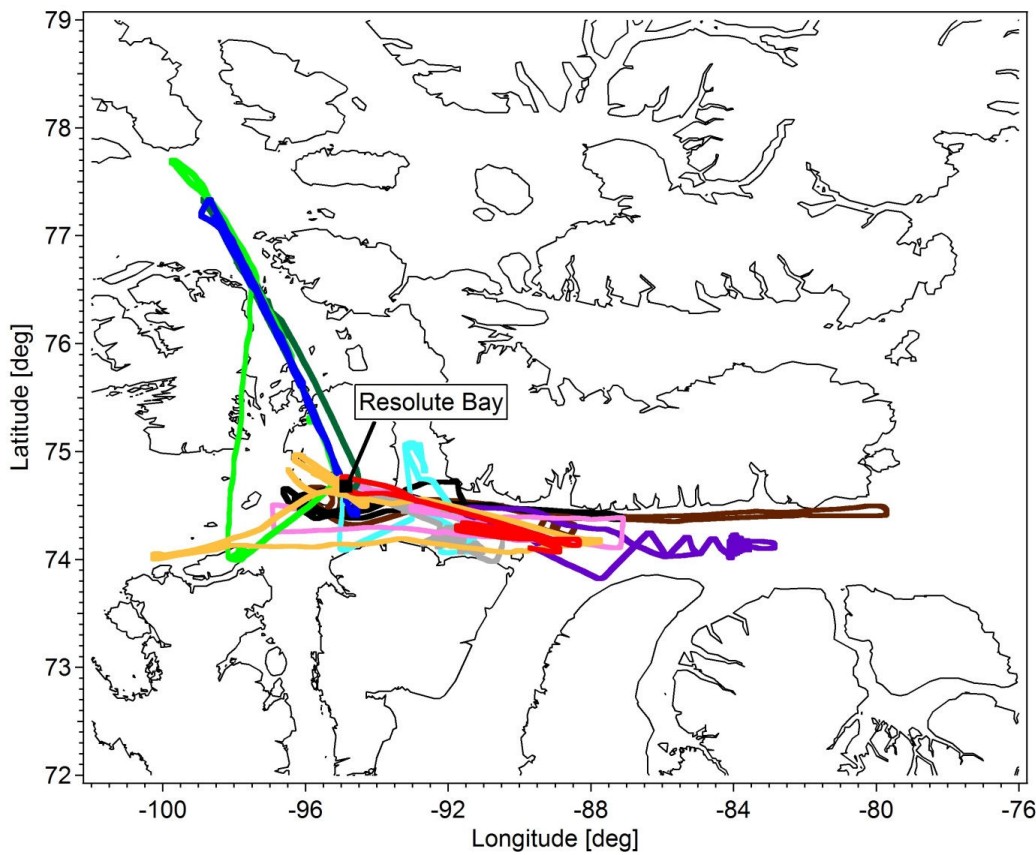

Figure 1. Compilation of the flight tracks. All flights originated from Resolute Bay (74°40'48"N 94°52'12"W).





July 5, 2014    July 8, 2014

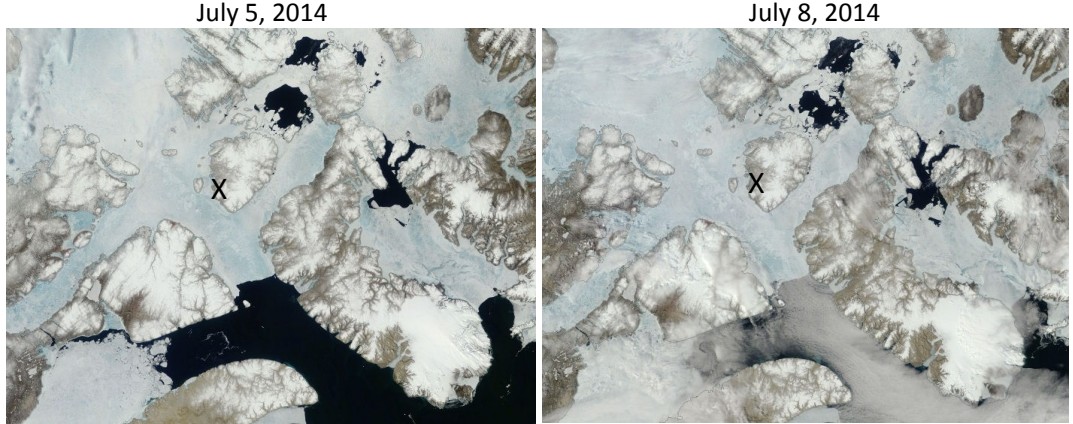

Figure 2. Satellite images from July 5 when LA clouds were sampled over the two polynyas to the north and from July 8 when LA clouds were sampled along Lancaster Sound (July 8). Lancaster Sound is cloud free on July 5 and mostly covered by cloud on July 8. Resolute Bay is marked with a "X". Images are courtesy of NASA Worldview: https://earthdata.nasa.gov/labs/worldview/.





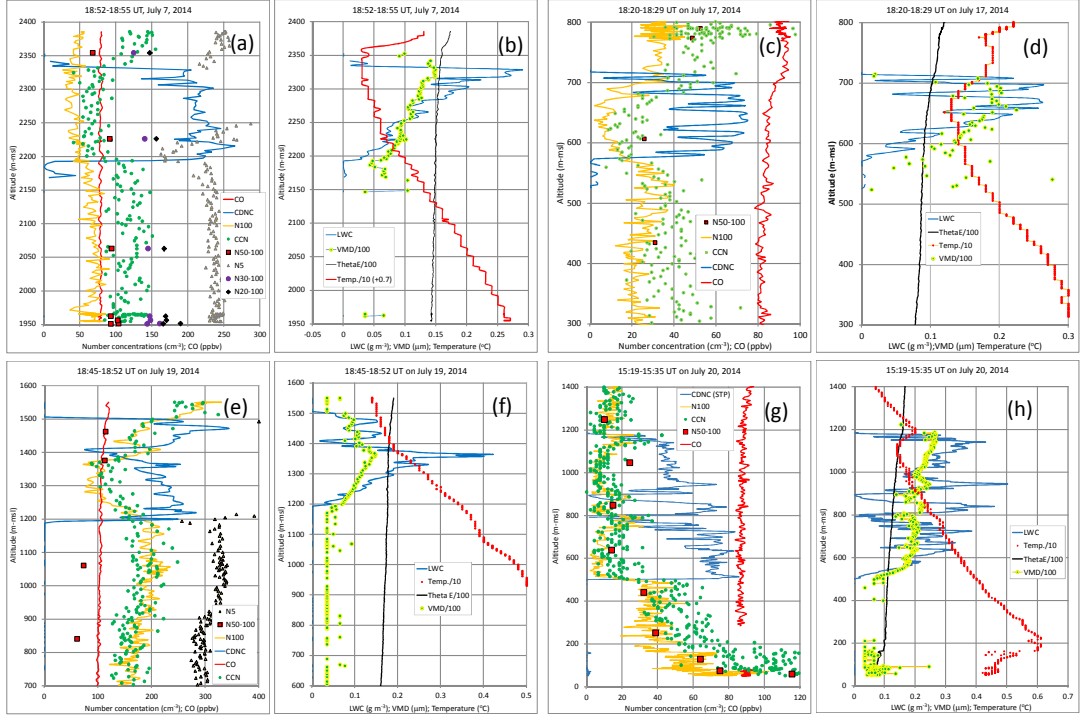

Figure 3. Four examples of profiles through HA clouds. a) Case from July 7 showing CO, CDNC, CCNC and particle number concentrations, where Nx-100, N100 and N5 are for particles sized between "x" nm and 100 nm, >100 nm and >5 nm respectively. b) Case from July 7 showing LWC, VMD, $\theta_e$ and temperature, where VMD, $\theta_e$ and temperature have been scaled as indicated in the legend. c) As in a), but case from July 17 and without N5. d) As in b), but case from July 17. e) As in a), but case from July 19. f) As in b) but case from July 19. g) As in a) but case from July 20 and without N5. H) as in b), but case from July 20. The CDNC are all referenced to STP, and $\theta_e$ is given in degrees Centigrade before scaling.



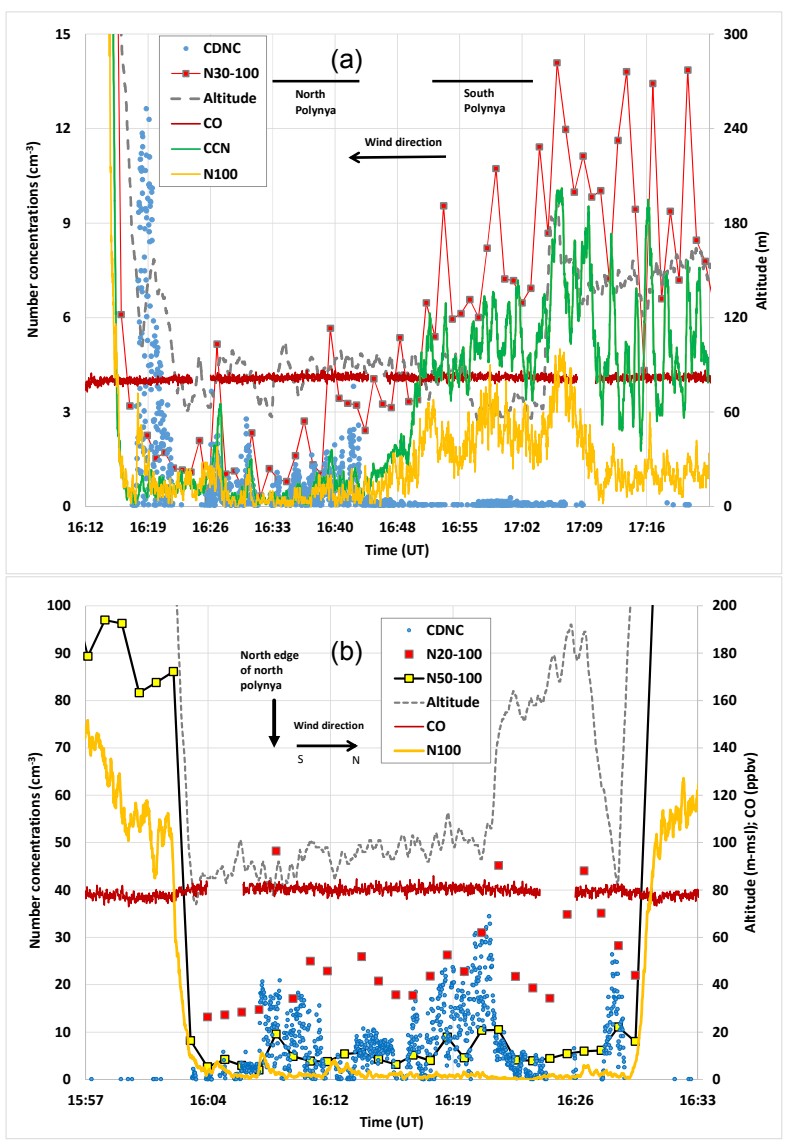

Figure 4.  Time series during the sampling of low (LA) cloud or fog over the polynyas north of Resolute Bay. a) July 5 time series showing CO, CDNC, CCNC and particle number concentrations, where N30-100 is for particles sized between 30 nm and 100 nm and N100 is for particles sized >100 nm. b) July 7 time series showing CO, CDNC and particle number concentrations, where N20-100, N50-100 and N100 are for particles sized between 20 nm and 100 nm, between 50 nm and 100 nm and >100 nm respectively. CCNC measurements are unavailable for this period on July 7. Wind direction and relative position of polynyas are indicated in both panels. CDNC are referenced to STP.





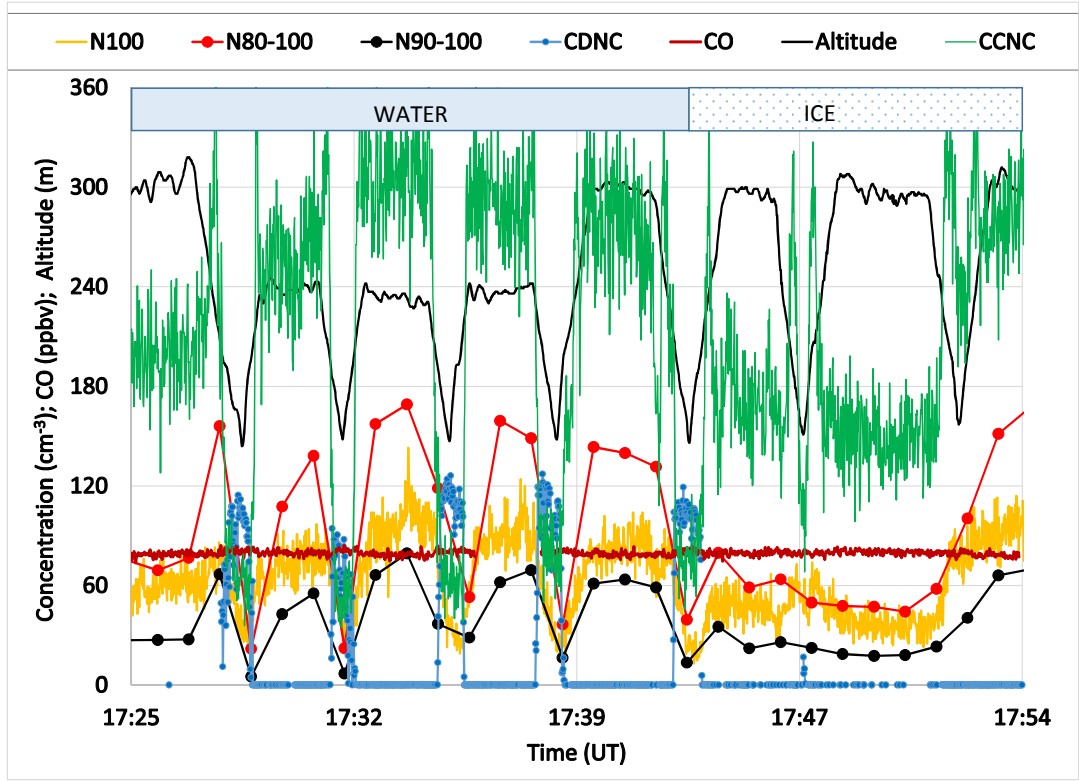

Figure 5. Time series of altitude, CO, N80-100, N90-100, N100, CCNC and CDNC from low cloud (LA) cloud sampling over Lancaster Sound on July 8. The cloud was deeper over the open water of the Sound (see satellite picture in Fig. 2b). Over the ice to the west, the cloud was not as deep and could not be sampled. Segments over water and ice are indicated at the top of the figure.





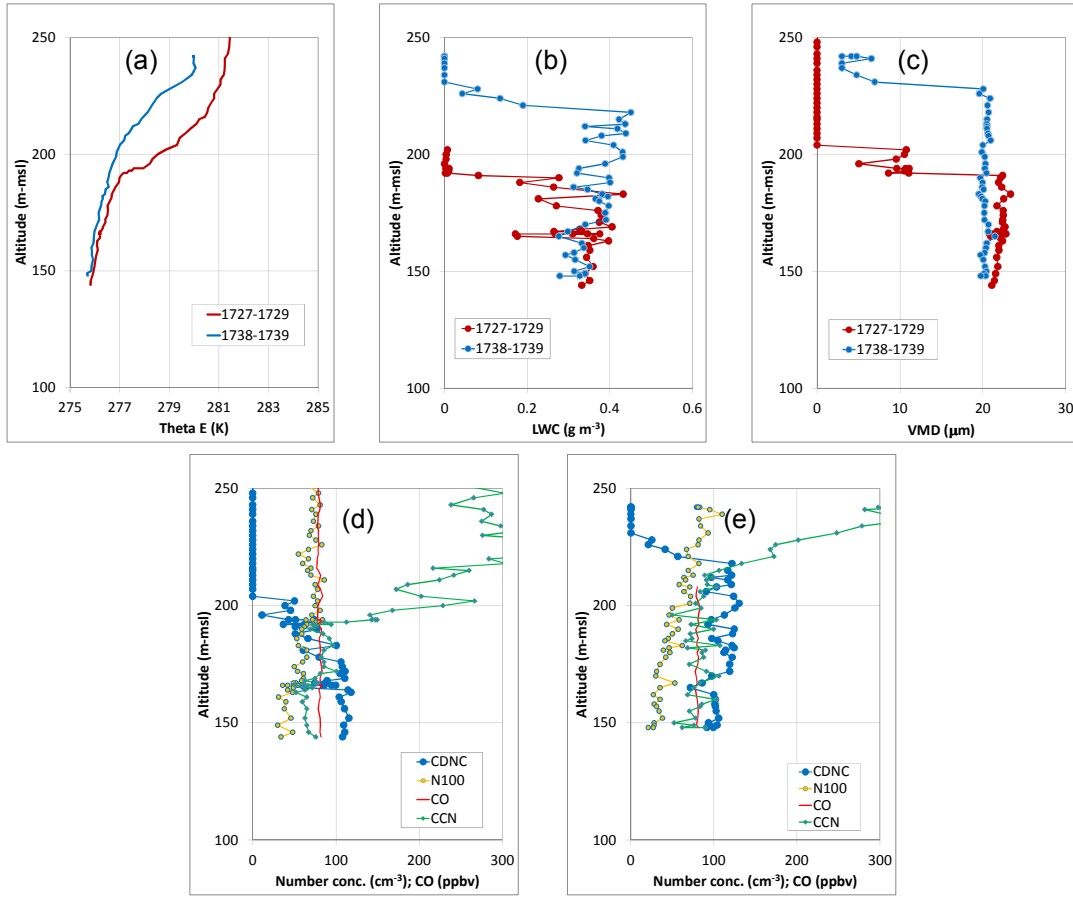

Figure 6. Profiles down into cloud showing a) θe, b) LWC and c) VMDData for periods 17:27-17:29 UT and 17:38-17:39 UT during July 8. d) shows CDNC, N100, CO and CCNC for the 17:27-17:29 UT profile, and e) shows CDNC, N100, CO and CCNC for the 17:38-17:39 UT profile.





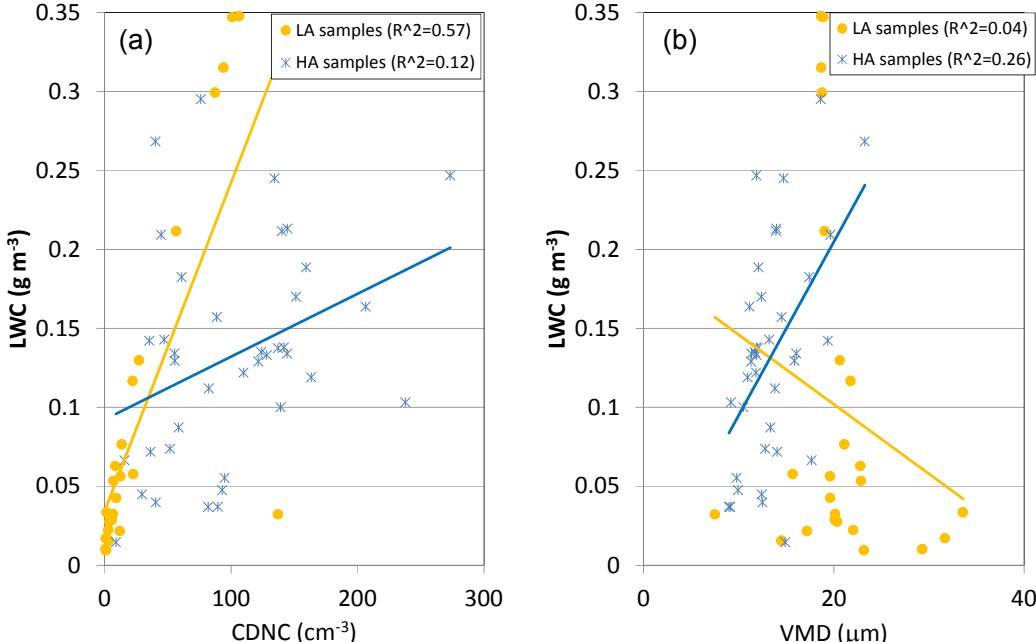

Figure 7. The LWC plotted as a function of the CDNC (a) and VMD (b) for the LA (orange) and HA (blue) samples. Linear regressions for each of the LA and HA samples are also plotted, and the coefficients of determination are given in the legends.




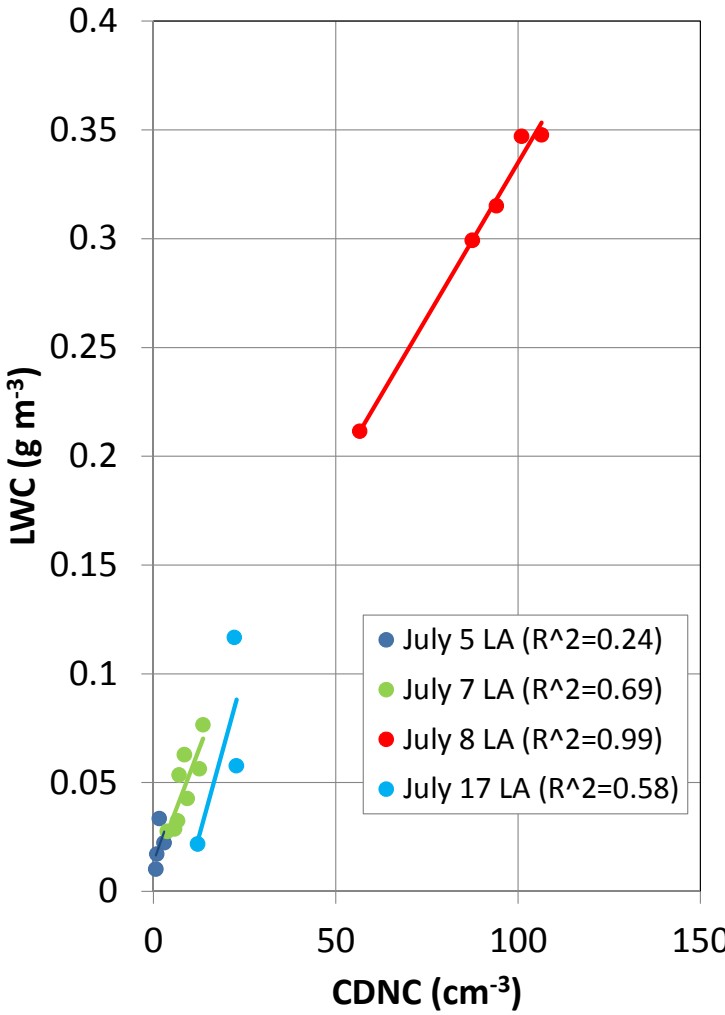

Figure 8. As in Fig. 7a, but only for four identified LA cases (July 5, 7, 8 and 11). Linear regressions for each set of samples are also plotted, and the coefficients of determination are given in the legends.







Figure 9. Plots of CDNC versus a) N100 and b) N50. Points are identified between LA (yellow) and HA (blue) samples, and the 1:1 lines are for reference.




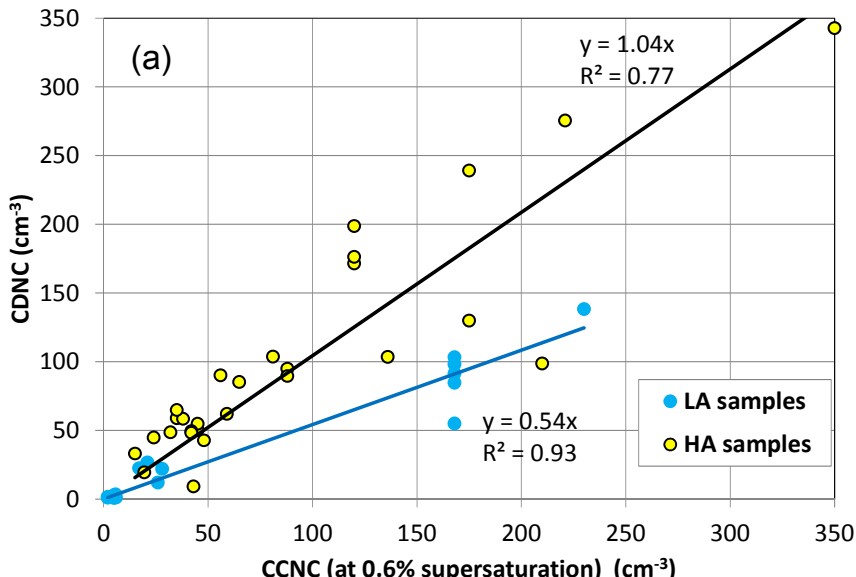

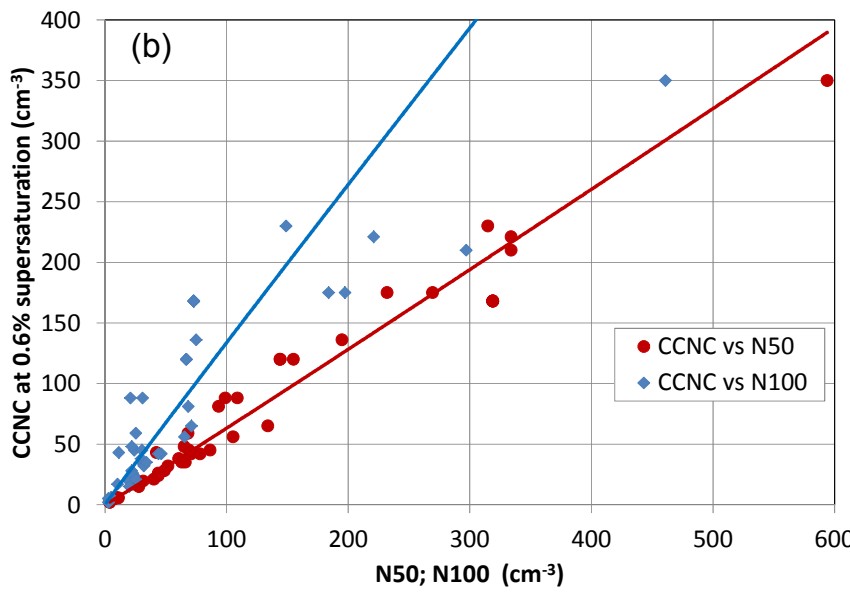

Figure 10. a) CDNC plotted versus the CCNC measured at 0.6% supersaturation; points are identified between LA (yellow) and HA (blue) samples, and linear regressions through the origin are shown. b) CCNC plotted versus N50 and N100; power law fits to each are provided for reference.




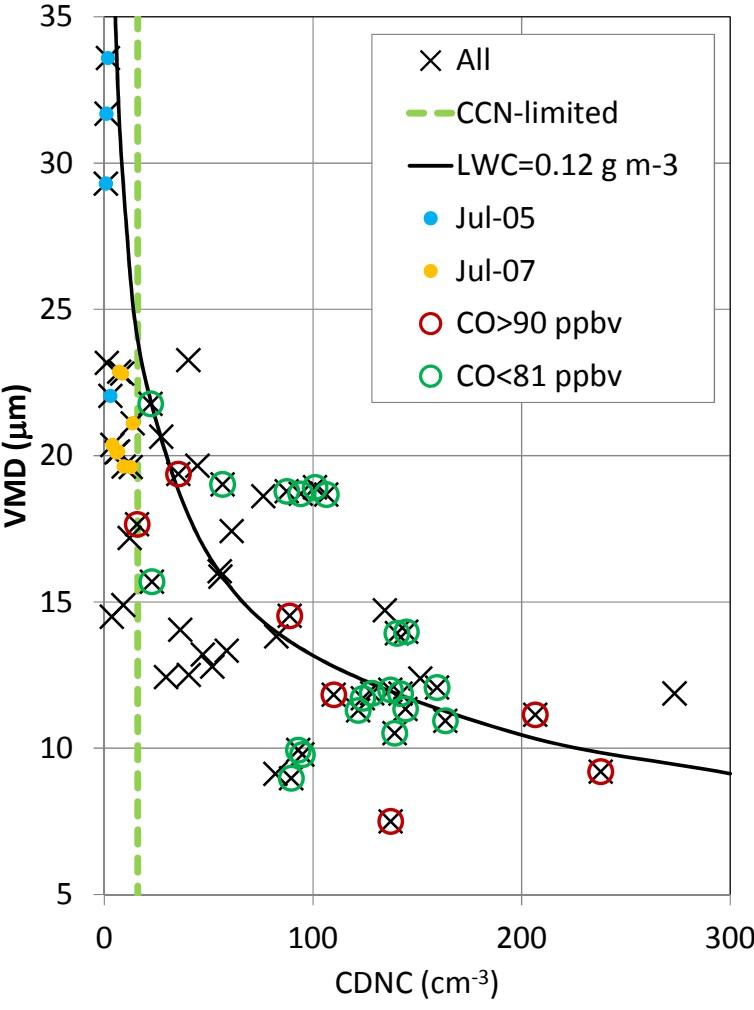

Figure 11. The mean VMD of all cloud samples plotted versus the CDNC. All CDNC are referenced to the in-situ pressure. The dashed vertical green line represents the "CCN-limited" division discussed by Mauritsen et al (2011) and estimated here as 16 cm$^{-3}$. The solid black line is another reference showing the relationship between VMD and CDNC for a constant LWC: the study mean LWC of 0.12 g m$^{-3}$ (Table 1). Samples with higher CO (>90 ppbv) are identified by the open red circles. Also highlighted for the discussion are LA samples from July 5 (red dots) and July 7 (orange dots). The median CDNC are 1.3 cm$^{-3}$ and 7.8 cm$^{-3}$, for July 5 and 7, respectively; the N50 are 6 cm$^{-3}$ and 8.3 cm$^{-3}$ for July 5 and 7, respectively; the N100 are 3 cm$^{-3}$ and 2.2 cm$^{-3}$ for July 5 and July 7, respectively.