# Peer review of "Effects of 20-100 nanometre particles on liquid clouds in the clean summertime Arctic"

_Atmospheric Chemistry and Physics, 2015_

## Referee Comment (RC1) · J. Hudson (Referee) · 25 Feb 2016

This manuscript of aircraft field measurements in the Arctic summer ought to be published after numerous corrections and changes that I have listed.

CDNC is an awkward representation of cloud droplet concentrations. It is used by fewer authors because, for instance, $N_c$ is shorter. Some use $N_d$ but that should be reserved for drizzle drops. Or in this manuscript where only concentration at 0.6% is used $N_{0.6\%}$. CCN concentrations should be $N_{CCN}$. CDNC differs from CCNC by only one of 4 capital letters and D and C between C and N are not so easily distinguished, this is too confusing. I had to constantly reread to be sure of which one was referred. Supersaturation should be abbreviated S to save a lot of space.

The CCN-limited regime of Mauritsen needs further description. This is not the sort of well accepted concept that seems to be implied in this manuscript. It has apparently not been cited in other papers. If it has, then please cite. It is only in this original paper where it is not referred to as Mauritsen except in the author list. Most importantly, it is incorrect to say that there is no aerosol limitation. Within this regime there may not be a linear relationship (or any relationship) between $N_c$ and $N_{CCN}$ but the mere fact that $N_c$ is so low is because $N_{CCN}$ is that low. This cannot be dismissed as no aerosol effect below the Mauritsen limit. There seem to be two separate aspects. One is the apparent loss of linearity (or any relationship) between aerosol and droplets at low concentrations. The other seems to deal with long and perhaps short wave radiative differences between the regimes.

Yum and Hudson (2001), which is cited, showed strong $N_{CCN}$ vertical gradients with much lower $N_{CCN}$ at lower altitudes. This seems consistent with the lower $N_c$ of LA than HA clouds. They attributed the low low altitude $N_{CCN}$ to cloud scavenging because when there were no clouds low altitude $N_{CCN}$ was higher and the vertical gradient disappeared. This "scavenging" must have been due to coalescence among droplets that reduce $N_c$ and thus $N_{CCN}$. Brownian scavenging would also be at work but would reduce only the small interstitial particles that should not serve as very good CCN. Hudson et al. (2015; JGRA) demonstrates the effects of coalescence scavenging. Coalescence cannot and should not be so easily dismissed as it seems to be in this manuscript. Coalescence is probably even more active in warmer summer than spring. Thus, the low low-altitude $N_c$ and $N_{CCN}$ are not necessarily (probably not) due to low $N_{CCN}$ natural sources. Low $N_{CCN}$ is what is left after cloud scavenging. See also Wylie and Hudson (2002; JGR). Scavenging would be a reason that $N_c$ and $N_{CCN}$ would not be related. Furthermore, coalescence results in larger (lower $S_c$) CCN. Furthermore, droplet and aerosol measurements at such low values are more uncertain and thus it is difficult to dismiss relationships.

There seems to be a recurring error. If this is not an error much more explanation needs to be given to such counterintuitive results. Higher concentrations in HA clouds should be associated with lower cloud supersaturations (S) and activation of only larger particles. Lower concentrations of LA clouds should result in higher S and activation of smaller particles. The latter is stated a couple of times. But in several (at least 3) places it is stated that HA clouds make higher S than LA clouds. This seems contradictory to other statements in the manuscript. If this is somehow true then it is big news and requires much further explanation.

Hegg et al. (1995; JAM and 1996; JGR) and Radke et al. (1976; JAM) also made Arctic CCN measurements, the latter in June.

In the process of commenting on several manuscripts by Asian authors, whose native languages do not include articles, I have been forced to conclude that articles are often overused, especially the definite article. This is more of a tendency on this side of the Atlantic. Brits tend

to shun the more than Americans; i.e., "call police" or "go to hospital" rather than "call the police" or "go to the hospital".  Since in earlier decades I noticed that Canadians had British English tendencies I am surprised at the overuse of the definite article in this manuscript.  Perhaps this is Americanization.  Since continental Europeans would tend more toward British English and since this is a European journal it seems appropriate to cull the thes.  Moreover, when nearly every noun is preceded by the, the loses whatever impact the has.

L27.  Delete 2$^{nd}$ the.
L28.  Delete 2$^{nd}$ the.
L34.  Space before 50.
L35.  Higher concentrations for HA clouds should produce lower S.  Smaller particles are more likely associated with LA clouds.
L37-40.  Line 37, "(CCN)-limited regime of Mauritsen" and 38, "In that CCN-limited regime" are contradicted by line 40, "suggesting no aerosol limitation."  Just what does CCN-limited mean?  Why is this Mauritsen regime called CCN-limited?  Within this regime the CCN do not appear to cause any limitation because $N_c$ is not related to $N_{CCN}$.
L48.  Delete the.
L49.  Delete 2$^{nd}$ the.
L52.  Delete 2$^{nd}$ the.  Change last the to a.
L53.  Delete last the.
L59.  Delete last the.
L59-61.  Is this a trend over time or a constant factor?
L68.  Delete the.
L77.  Hudson et al. (2010) could be cited here as well.
L81.  Insert that before may.
L82-5.  This sentence is too long and thus too confusing.
L83.  Delete –based.
L90-1.  Not clear.  How is this required?
L98.  Insert to be after considered.
L100.  Delete 2$^{nd}$ , 3$^{rd}$ and 4$^{th}$ the.  Delete 2$^{nd}$ of.  Move effects after aerosol.
L112.  Delete the.  Concentration plural.
L114.  Delete the twice.  Delete within.  Delete , and.  Move cloud in front of microphysics.  Period after microphysics.
L115.  Insert Moreover, .  concentration plural.  Was to were.
L131.  Delete are.  Delete ing.  Period after droplets.  Change which will to This would.
L156.  Delete nm.  Change 1 µm to 1000 nm.
L161.  Is this the internal pressure of the instrument?  If so then so note.
L191-3.  Say that vertical wind could not be measured rather than this round-about statement.
L202.  Delete at. Insert of after off.
L219.  Could this be intense.
L229-30.  For the entire project or period 2?
L250.  Change The latter to Descent.
L252.  Delete based on.
L254.  Delete 2$^{nd}$ and 3$^{rd}$ the.
L274-6.  Seems to be adiabatic?
L276.  What is the alternate cloud formation?

L279.  Is this not nucleation, which should not be called scavenging?
L281-2.  Citation.
L286-8.  Closure of what concentrations.  More explanation.
L288.  Either values or compares need to lose s.
L289.  Why?
291-2.  Depends on composition.
L294.  This comparison seems out of context.  Is there a reference for this?  If this is so important it should not be just supplementary.  These Atlantic measurements should have the same scrutiny as the Arctic measurements.
L302.  There seems to be more variability at all altitudes.  Delete the twice.  Move overall right after and.
L327.  Delete $2^{nd}$ the.  Add s to equal.
L329.  Insert apparently after were.
L332.  Delete the twice.
L346-7.  Explain this transfer.
L351.  In to within.
L363.  Reached low or high?
L368.  Delete $2^{nd}$ the.  Delete be.
L372.  Why/how is this characteristic of vertical mixing?
L396.  Apparently this is the cloud threshold used in this study.  This needs to be stated clearly.  This is a rather low threshold, many use 0.03 or 0.1.
L397.  Delete first seconds.
L398.  How about a standard deviation.
L402.  In-situ volume apparently means ambient volume.  STP is apparently in parentheses?  So state.
L411.  For above 200 m is awkward.
L413.  Figs.
L414.  To above 25 is awkward.
L415-6.  Can't this be more precise?
L427-8.  Needs citation.
L428-9.  Why would vertical motions not be responsible for clouds?
L432.  Delete the.
L440.  $R^2$ does not match Fig. 7a.
L444.  What is they?
L445.  Delete the thrice.  Delete by evaporation of droplets.  That phrase seems to sort of contradict without reducing $N_c$.
L447.  Rate of cooling with altitude?  Greater vertical wind (W)?  But this will not affect LWC, which depends on distance from cloud base not W.
L448-51.  Perhaps increased $N_c$ will reduce coalescence but not increased LWC.
L451.  This is not really an increase but rather higher LWC than otherwise.
L453-4.  Eliminate commas.  Change which to that.  This is a restrictive clause rather than a nonrestrictive clause.
L460-1.  Explain.
L461.  Relationship plural.  Insert is after it.
L461-3.  Why?
L462.  Um to µm.

L463.  Fig. 8 does not have panels.

L482-3.  Of course they do not.

L483.  Delete necessarily.  $2^{nd}$ to to of.

L488.  Less surface area.

L501-5.  This appears to be stated incorrectly, backwards.  If not much further explanation is required for these extremely unusual results.

L525-6.  How and why?

L536.  Insert below 16 after points.

L537.  Last the to these.

L542.  Delete the twice.

L543.  Comma after clouds.

L544.  Insert that before the.

L546.  Delete $1^{st}$ the.  Change $2^{nd}$ the to that.

L544-7.  Which differences are referred to here?  If it is among the July 5 measurements then ok.  But otherwise not.

L555-6.  Has this been demonstrated previously?  If so cite.  I do not seew how you can discount coalescence.  Or why this is needed.

L558.  Must be referring to vertical wind?  So state.

L565.  Aerosol impact is ubiquitous.  $N_c$ below Mauritsen is caused by low $N_{CCN}$.  You cannot get around this.

L590-1.  Again this is the cloud threshold and should be so stated.

L605.  And to with.

L606.  LA clouds with lower $N_c$ should have higher not lower S.

L609.  Insert relatively clean after for.

L621-2.  All samples exhibit a clear influence of the aerosol.  There are differences between the two regimes.

L623.  Not sure you can call them polluted.

L625.  There is no proof of natural sources.

Table 1.  Explain parentheses.  Apparently no distinction of LA and HA?

Table 2.  got to at?

Most figures need tic marks.

When there are various panels there should not be repetition of every axis label and title.  This correction could help allow larger panels.  Many panels are small and difficult to read.

Fig. 3.  Is a prime example.  It is not necessary to write the horizontal axis titles for a-d as they are the same as the panels below.  The horizontal axis ranges could also be adjusted so that it is not necessary to write the labels for a-d as they can be the same as those below.  B, d,f and h do not need vertical axis titles and labels as they are the same as a, c, e and g respectively.

Fig. 6 should be landscape with panels side-by-side and then only one vertical axis title and label would be necessary for a.  Panels should then be moved closer to each other.  Moreover, the figures could be larger.

Fig. 7  Vertical axis titles and labels for b should be removed and panels moved closer together.  Orange appears yellow.

Fig. 8.  Is it July 17 or 11?  With so few data points probability or significance levels should be shown.  This could also be done for Fig. 7.

Fig. 9.  Rescale horizontal for a (0-700) and remove labels.   Or make landscape with panels side-by-side with the same vertical scales thus removing b labels.  There are no blue data points and yellow seems to be mislabeled.

Fig. 10.  Reverse the axes of b so that CCN is the horizontal for both and then the horizontal title and label can be removed from panel a.  Or reverse axes of a and plot side-by-side using landscape.

Fig. 11 caption.  In-situ probably means ambient.  July 7 appears yellow not orange.  July 5 is blue not red.

---

## Referee Comment (RC2) · Anonymous Referee #2 · 26 Feb 2016

Anonymous Review, doi:10.5194/acp-2015-999, Leaitch et al., 2016 ACPD

**General comments:**

The topic of this manuscript is very interesting and worthy of study. The authors provide information on clean liquid clouds in the Arctic that will help us more fully understand the Arctic radiative system, particularly during the under-studied, cleaner summertime environments. The study improves our mechanistic understanding of how aerosols affect these clouds and it provides a needed basis for understanding how changes from anthropogenic aerosols affect the Arctic. This is a topic of relevance and importance. In principle, I think this manuscript may be worthy of publication. However, there are a few changes and clarifications that I would like to see before finally recommending acceptance for publication.

First, the paper could really benefit from a more complete discussion of uncertainties and, in some places, a more thorough statistical analysis. I suggest the authors more fully a) substantiate of some of the claims, b) compare to findings from other works, and c) clarify what is known, not known, and what is just a best guess. The section that was most problematic was the discussion of their methods and assumptions regarding calculating activation size from aerosol concentrations and CCN. I provide more detailed comments and suggestions in the specific comments below.

Secondly, the paper would be more useful to the community if the writing and organization were altered to provide more clarity. I was frequently confused throughout the manuscript about what the authors were doing and why, and I had to re-read some parts multiple times over to understand those parts. Some parts of the manuscript could benefit from more selectively discussing relevant information and eliminating irrelevant information; other parts of the paper would be improved by a more detailed explanation of why the reader should be interested. It might be helpful if the authors could consolidate some of the written information into useful figures or tables. There were also, in my opinion, a high level of references to the supplementary material – to the point that the manuscript does not stand well on it's own. I have given more specific comments and suggestions regarding these points below.

**Specific comments**

*Most important comments:*

1) In the introduction, the authors argue that particles in the 20-100 nm diameter range can become CCN in low aerosol environments. As best I can tell, outside this paper, their initial argument is based only on one previous study (Leaitch et al., 2013). Since a) globally, environments with such low aerosol are fairly rare, b) this hypothesis is not commonly known of or accepted in the community, and c) this hypothesis is a fundamental question/argument to their paper, the authors should describe their reasoning for this argument more fully. For example, if

available, the authors should provide more literature references to support the existence of this process. It would also be helpful to expand on why specifically the findings of the Leaitch et al. (2013) paper support the author's hypothesis. Adding this context will make it easier to evaluate the work.

2) In this paper, the authors make the case that small particles do nucleate cloud droplets based on the following assumptions. First, they assume that the cloud is only incorporating aerosols from below the cloud. Second, they assume that aerosols are preferentially nucleated by size, with larger particles being nucleated first. Thus, they estimate the activation sizes of CCN from the number of CCN in-cloud and the number of aerosols below-cloud in various size classes. However, the various uncertainties in these assumptions need further discussion. What about the impacts of chemistry and hygroscopicity of the individual particles? What about in-cloud aerosol processing or entrainment from above or from the sides? What about precipitation from above? (was there any? Multi-layer clouds are very common in the Arctic). A more clear and straightforward introduction into their assumptions and the uncertainties in these assumptions would be helpful.

3) Other uncertainties that should be discussed more:

   a. Impacts of meteorology on the results. Section 2.3 describes (in way too much detail, I think) the conditions of the study. A lot of information is presented there, but much of it is not directly relevant to evaluation of the key points of the manuscript. At the same time, some very important information is missing, like cloud types (e.g., stratus, cumulus, etc.), cloud sizes, lifetimes, and whether precipitation was occurring. In the summary and conclusions section, it was briefly mentioned that some of the clouds were stratus clouds, but there was no mention of how many. This information should be added sooner and with more detail. There should also be a more thorough discussion of how meteorology and/or surface conditions might or might not affect the results.

   b. Reliability of in-cloud measurements. Were the in-cloud CCN values reliable? What about the aerosol measurements (I know the authors said these were not used in the analysis, but they are shown in Fig. 3)? From what I understand, lot of instruments do not function as well in-cloud as out-of-cloud.

   c. The ability to adequately characterize the influence of smoke from the CO and aerosol data alone (e.g., P.27, l. 595). On P. 14, l. 311 Koellner et al., 2015 is referenced to say that the very slightly elevated CO was from BB. However, I do not think it is appropriate to cite a non-peer reviewed conference abstract to make a point here. Also, CO levels of 100+ ppbv are still quite low compared to other measured CO values in the Arctic, and other summertime studies still count values of CO above this level as "background air" (e.g., Sakamoto et al. (2015, doi:10.5194/acp-15-1633-2015)). Also, particle concentrations were still quite low, weren't they? The authors might consider restating this sentence to mention that the CO levels were elevated compared to your other days, perhaps due to *dilute* BB, but that they were still pretty clean. From what I understand, photochemical destruction of CO would lower average CO values in the summer compared to other seasons, making it

additionally difficult to compare aerosol:CO values to assess influence from biomass burning smoke. Also, there are other source of CO and aerosols besides smoke, and being a gas, CO is an imperfect smoke aerosol tracer to begin with. Anyways, I think the claim that these air masses were influenced by smoke is more uncertain than is currently presented.

    d. P. 24, l. 536: there is an awful lot of scatter (understandably) in figure 11; thus I think there should be some discussion of the uncertainties of the 16 cm$^{-3}$ limit the authors use as their estimate of the Mauritsen limit. Relatedly, on this line I would not call it "the Mauritsen limit", but rather, "our best-guess Mauritsen limit."

4) Other key information that was missing or unclear:

    a. In the introduction/methods there was not a clear, concise description of how the questions would be tested, and how the methods would provide useful information. Please add.

    b. Given that 62 different clouds were examined, it was not clear to me why a select few were chosen for more detailed analysis. Please provide this information.

    c. Why were LA and HA clouds specifically separated at 200 m?

    d. p. 13, l. 292: "for 20 nm particles to activate, the water supersaturations in the bases of the clouds would have had to reach about 1.5%." How was this calculated?

5) There is a heavy reliance on the supplementary figures, and it makes it harder on the reader to have to download another document and scroll through just to understand the main text. It would be better to just put the relevant figures in the main text.

6) I think the authors are actually underselling a couple things:

    a. the fact that no one has observed this Mauritsen hypothesis in the field before. This point makes their work more exciting and I think they could emphasize it more.

    b. the fact that the clouds they are describing are so clean. As noted in the general comments, a better baseline of clean clouds provides a much-needed basis for understanding how changes from anthropogenic aerosols affect the Arctic and this lends importance to the study. This point could also be emphasized more.

*Other specific comments:*

7) Statements that should be backed up more with further statistical analysis:

    a. p. 14, l. 304: "The VMD exceed 20 um compared with a peak value of about 15 um for the July 7 profile." Is this statistically significant difference? Also, please give more information on why this statement is relevant to the paper.

    b. P.22, l. 491: where did you get the 40% value from? Is this statistically relevant?

    c. P.23 l. 503: you say there is a significant difference. What methods did you use to determine the significance, and how significant is it?

8) Other missing or unclear information:

a. P.24, l. 524: "the vertical dashed green line represents the Mauritsen limit below which the cloud may produce a net warming for an increase in the CDNC. " 1) in the earlier text, this limit was estimated as 10 cm$^{-3}$, and in the figure it is estimated at 16 cm$^{-3}$. Since you bring up the figure here first, can you please describe here why there is this difference, and not on line 535, where it is mentioned first? 2) Also, please state why you say in the above text that below this line the cloud can produce net warming. Are you relying on the Mauritsen model results for this statement? If so, please say so.

b. P25, l.558-560: "Variations in other processes, such as mixing or the rate of cooling, may be responsible for the correlation of CDNC and LWC. In these cases, there is sufficient aerosol so as not to be a limiting factor for the CDNC." How do you know that?

c. P.26L l. 577: "The lower- and higher-CO points overlap over a CDNC range of 16 cm$^{-3}$ to 160 cm$^{-3}$." The way this is phrased is a little confusing. I am not sure how it tells you "that 20-100 nm particles from natural sources can have a broad impact on the range of CDNC in clean environments." (l. 579-580)

d. P, 4, l. 81, "remote sensing particle Na do not reflect the size distribution" can you reword to make this sentence clearer, e.g., small particles are not observable, or something like that? I think this could be a very important point for your argument, so it is important to make sure it is clear. (also, please add a reference)

e. P. 11, l. 236: can you explain what you mean by saying that clouds were sampled more frequently during period 1 because of "better visual contrast"? I am confused.

f. Lines 292-300: It is not really explained why the authors are comparing and contrasting the July 7 case with some other unrelated cloud case. Can this be made more clear, or taken out?

9) Abstract: The authors might consider adding a sentence on what this all means for the bigger picture.

10) I agree with Dr. Hudson that the authors should consider using a different to describe cloud droplet number concentration and concentration for cloud condensation nuclei concentration. Currently, the authors use CDNC and CCNC, respectively, and these abbreviations look quite similar, which could lead to some confusion.

11) p. 13, l. 288: "The N30 values compares most closely with the mean CDNC…" [I did not see that in Fig. 3a?]

12) p. 11, l. 246, why are all Na concentrations for particles < 100 nm diameter taken from the SMS, when you also have TSI data down to 5 nm from a different instrument?

13) Did SMS and UHSAS match up fairly well in their overlapping section and generally speaking? Because each might have different artifacts, do you have evidence that these artifacts are either small, or at least similar?

14) p. 12 l. 272: you say it is one of several profiles through the same cloud. How do the other profiles compare?

15) P. 16, l. 351: "The CO mixing ratio is slightly higher in the cloud (81 ppbv) than above (79 ppbv)." I doubt that is statistically significant and outside the error of the instrument. I suggest taking this sentence out.

16) Fig. 8: Are each the points the mean (or median, or something like that) of an individual cloud? Can that be stated more clearly in the caption? Why are clouds within each flight considered separately from each other?

17) p. 21, l. 463: There is no figure 8a. Did you mean Fig. 8? If so, can you be more clear about why LWC and VMD indicate anything for Fig. 8?

*Technical remarks*

1) Fig. 1: can you add an inset that shows Canada, and where Resolute Bay is relative to the larger region? It would also be helpful to show which areas are land vs. ocean.

2) It would be useful to edit Section 3.4.1 for clarity (it is currently 13-sentence paragraph, so breaking it up would help).

3) Section 2.4.1 could be substantially shortened, and made into a table or figure to better elucidate the main points the author wishes to get across. One suggestion would be to add this information on supposed activation sizes into Fig. 3.

4) p. 11, l. 242: "Number concentrations [of particles] greater than 100 nm"?

5) p. 13, l. 289: by, "in the mean " do you mean "on average"?

6) p. 14, l. 302: more variability in what specifically?

7) Fig. 4a: is CO the same axis as altitude, as in Fig. 4b?

8) Fig. S7: Caption says LWC and VMD should be shown, but I only see CDNC

9) Fig. 9: caption says blue, but the points are yellow and black symbols.

10) P22, l48: "The CDNC are plotted against the N50 in Fig. 9b showing that the mean activation size [of HA clouds] was often close to 50 nm."

11) Figure 10: what is this? All the samples? Caption and discussion in the text need to be more clear.

12) Figure 11: you refer to red dots, but I think you meant blue dots

13) Figure S8: do these points represent averages of single clouds? Please specify.

14) P.25, l. 554: "In this low CDNC environment, where cloud droplets may grow large enough to be gravitationally removed from the cloud without the support of collision-coalescence." Reference?

15) P. 25, l.: The reader is referred to the Fig. 11 caption. This information would be more appropriate for a table or for the text.

*16)* P.26, l. 577: The reference of Burkart et al., 2015 is another conference abstract and is not a suitable reference.

17) P. 16, l. 355: "particles [of] about 50 nm"?

18) Table 1 and 2 can be combined into 1 table.

19) P21, l. 475: It is stated that, "As shown in the plot of CDNC versus N100 (Fig. 9a), particles **smaller than** 100 nm activated in most cases", but on p.11 it is said that, "Number concentrations **greater than** 100 nm (N100) are taken from the UHSAS." So on line 475, instead of "smaller than", did you mean "greater than"?

20) P. 27, l. 612: I'd suggest saying ~16 instead of simply 16.

---

## Author Comment (AC1) · 10 May 2016

**Responses to the Reviewers.**

We thank both Reviewers for their thorough comments that have improved this paper.

Reviewer comments are highlighted in yellow, and responses follow. The line numbers refer to the revised manuscript.

**Reviewer 1 (Dr. Hudson)**

This manuscript of aircraft field measurements in the Arctic summer ought to be published after numerous corrections and changes that I have listed.

Response - We are very grateful to Dr. Hudson for his careful and detailed consideration of this work.

1) CDNC is an awkward representation of cloud droplet concentrations. It is used by fewer authors because, for instance, Nc is shorter. Some use Nd but that should be reserved for drizzle drops. Or in this manuscript where only concentration at 0.6% is used N0.6%. CCN concentrations should be NCCN. CDNC differs from CCNC by only one of 4 capital letters and D and C between C and N are not so easily distinguished, this is too confusing. I had to constantly reread to be sure of which one was referred. Supersaturation should be abbreviated S to save a lot of space.

Response – We have not changed CDNC as it is descriptive and maintains a more consistent approach to all of the abbreviations in this case. In order to avoid confusion and provide more clarity, we have changed CCNC to CCNC(0.6%).  Further, we have abbreviated supersaturation to "S" as suggested.

2) The CCN-limited regime of Mauritsen needs further description. This is not the sort of well accepted concept that seems to be implied in this manuscript. It has apparently not been cited in other papers. If it has, then please cite. It is only in this original paper where it is not referred to as Mauritsen except in the author list. Most importantly, it is incorrect to say that there is no aerosol limitation. Within this regime there may not be a linear relationship (or any relationship) between Nc and NCCN but the mere fact that Nc is so low is because NCCN is that low. This cannot be dismissed as no aerosol effect below the Mauritsen limit. There seem to be two separate aspects. One is the apparent loss of linearity (or any relationship) between aerosol and droplets at low concentrations. The other seems to deal with long and perhaps short wave radiative differences between the regimes.

Response – The CCN-limited regime is documented in Mauritsen et al. (2011). We have discussed the nature and consequence of the limit, but the details can be found in the reference. As Reviewer 2 suggests, it seems that our data set is the only one currently available to address in some way the relationship among the aerosol, CDNC and LWC below the Mauritsen limit. That is one explanation for the apparent lack of general acceptance of this regime.  We agree that we cannot say there is no aerosol limitation. Our data only suggest the absence of change in the CDNC and LWC associated with change in the aerosol. We acknowledge your point that a reduction of the aerosol has led to this regime, but we are focused on what controls the CDNC and LWC within this regime. On lines 38-40 of the abstract, we have revised the text to read "These first observations of that CCN-limited regime indicate a positive association of the liquid water contents (LWC) and CDNC, but no association of either the CDNC or LWC with changes in the aerosol." On lines 602-604, we now state "It can be argued that some aerosol must

exist for these clouds to form, but these observations show no association of changes in either the CDNC or LWC with changes in the aerosol." Finally, on lines 665-667 of the conclusions, we now state "These observations show no association of changes in either the CDNC or LWC with changes in the aerosol within the Mauritsen limit."

3) Yum and Hudson (2001), which is cited, showed strong NCCN vertical gradients with much lower NCCN at lower altitudes. This seems consistent with the lower Nc of LA than HA clouds. They attributed the low low altitude NCCN to cloud scavenging because when there were no clouds low altitude NCCN was higher and the vertical gradient disappeared. This "scavenging" must have been due to coalescence among droplets that reduce Nc and thus NCCN. Brownian scavenging would also be at work but would reduce only the small interstitial particles that should not serve as very good CCN. Hudson et al. (2015; JGRA) demonstrates the effects of coalescence scavenging. Coalescence cannot and should not be so easily dismissed as it seems to be in this manuscript. Coalescence is probably even more active in warmer summer than spring. Thus, the low low-altitude Nc and NCCN are not necessarily (probably not) due to low NCCN natural sources. Low NCCN is what is left after cloud scavenging. See also Wylie and Hudson (2002; JGR). Scavenging would be a reason that Nc and NCCN would not be related. Furthermore, coalescence results in larger (lower Sc) CCN. Furthermore, droplet and aerosol measurements at such low values are more uncertain and thus it is difficult to dismiss relationships.

Response – There is no doubt that scavenging contributed to the cleanliness of the atmosphere in this study. The presence of cloud will reduce some of the smaller interstitial particles, and some particles can be reduced by rainout and washout. We don't see evidence for larger CCN as reflected in the N100. Also we have evidence (Willis et al., ACPD, 2016, and other work in preparation) that ultrafine particles are replenished in the very clean air in the absence of cloud and above cloud. We cannot answer the question as to whether the scavenging that contributes to this relatively clean atmosphere results from the sampled clouds or whether it due to past cloud events. We don't dismiss collision-coalescence (C-C), but when the observations indicate that you have between 1 and 5 droplets per cc that are 20-30 $\mu$m in diameter, it is hard to expect C-C to be significant. It is possible that C-C was a factor in leading up to that point, but we have no observations to help with that. The other factor is that these high VMD, low CDNC clouds (or fogs) were sampled over polynyas; the cloud or fog was formed by moisture from the polynyas cooling in the air above under relatively low wind conditions. So these are relatively isolated clouds that are unlikely to generate large numbers of activated droplets for any aerosol concentration.

4) There seems to be a recurring error. If this is not an error much more explanation needs to be given to such counterintuitive results. Higher concentrations in HA clouds should be associated with lower cloud supersaturations (S) and activation of only larger particles. Lower concentrations of LA clouds should result in higher S and activation of smaller particles. The latter is stated a couple of times. But in several (at least 3) places it is stated that HA clouds make higher S than LA clouds. This seems contradictory to other statements in the manuscript. If this is somehow true then it is big news and requires much further explanation.

Response – This is not an error. It stems partly from the very low concentrations of larger particles everywhere measured, but it is not only particle concentrations that control the S. The S is higher in the HA clouds (>200 m) because they have modest updrafts associated with them, whereas the LA clouds (<200 m) are near the surface and are more similar to fogs (if not fogs).  We have added some discussion to Section 2.3 for clarification: lines 252-259.

5) Hegg et al. (1995; JAM and 1996; JGR) and Radke et al. (1976; JAM) also made Arctic

CCN measurements, the latter in June.

Response – The Radke et al reference has been added: line 536.

In the process of commenting on several manuscripts by Asian authors, whose native languages do not include articles, I have been forced to conclude that articles are often overused, especially the definite article. This is more of a tendency on this side of the Atlantic. Brits tend to shun the more than Americans; i.e., "call police" or "go to hospital" rather than "call the police" or "go to the hospital". Since in earlier decades I noticed that Canadians had British English tendencies I am surprised at the overuse of the definite article in this manuscript. Perhaps this is Americanization. Since continental Europeans would tend more toward British English and since this is a European journal it seems appropriate to cull the thes. Moreover, when nearly every noun is preceded by the, the loses whatever impact the has.

Response – Thank you. We have considered each suggestion of the Reviewer and culled many of the "the"s. We have also made many of the various little changes as suggested by the Reviewer. We do not respond individually to those many specific comments below, but the changes are identified in the revised manuscript.

L27. Delete 2nd the.
L28. Delete 2nd the.
L34. Space before 50.
L35. Higher concentrations for HA clouds should produce lower S. Smaller particles are more likely associated with LA clouds.

Response – Refer to our response to the Reviewer's comment 4 above.

L37-40. Line 37, "(CCN)-limited regime of Mauritsen" and 38, "In that CCN-limited regime" are contradicted by line 40, "suggesting no aerosol limitation." Just what does CCN-limited mean? Why is this Mauritsen regime called CCN-limited? Within this regime the CCN do not appear to cause any limitation because Nc is not related to NCCN.

Response – This has been modified in response to both Reviewers. See also, our response to the Reviewer's comment 2 above.

L48. Delete the.
L49. Delete 2nd the.
L52. Delete 2nd the. Change last the to a.
L53. Delete last the.
L59. Delete last the.
L59-61. Is this a trend over time or a constant factor?

Response – The evaluation used climate models to consider observed temperature variations over a 90-year period. It includes GHGs as well as aerosols. Nothing is constant over that time.

L68. Delete the.
L77. Hudson et al. (2010) could be cited here as well.

Response – Added: line 83.

L81. Insert that before may.

Response – Inserted.

L82-5. This sentence is too long and thus too confusing.
L83. Delete –based.

Response – This segment has been re-written: lines 73-86.

L90-1. Not clear. How is this required?

Response – The sentence has been re-written: lines 90-91.

L98. Insert to be after considered.

Response – "considered" removed.

L100. Delete 2nd , 3rd and 4th the. Delete 2nd of. Move effects after aerosol.
L112. Delete the. Concentration plural.
L114. Delete the twice. Delete within. Delete , and. Move cloud in front of microphysics. Period after microphysics.
L115. Insert Moreover, . concentration plural. Was to were.
L131. Delete are. Delete ing. Period after droplets. Change which will to This would.
L156. Delete nm. Change 1 μm to 1000 nm.

Responses: These corrections have been made.

L161. Is this the internal pressure of the instrument? If so then so note.

Response – It is stated that this is the pressure in the inlet to the CCN instrument, which means that is also the internal pressure in the chamber.

L191-3. Say that vertical wind could not be measured rather than this round-about statement.

Response – The statement is correct in reference to gusts. These instruments do not measure broad scale vertical winds.

L202. Delete at. Insert of after off.

Response – Done: line 202.

L219. Could this be intense.

Response – intensive is removed.

L229-30. For the entire project or period 2?

Response – For the study period. Sentence re-written: lines 228-230.

L250. Change The latter to Descent.

Response – Re-written: lines 252-259.

L252. Delete based on.

Response – Not changed.

L254. Delete 2nd and 3rd the.
L274-6. Seems to be adiabatic?

Response – It does appear to be close to adiabatic, except near the top. We now mention this on line 320.

L276. What is the alternate cloud formation?

Response – There has been discussion that entrainment from above cloud, or mixing from the top down, can influence CDNC. There may be some possibility of radiative cooling as the air is transported.

L279. Is this not nucleation, which should not be called scavenging?

Response – This has been changed to "nucleation scavenging": line 302.

L281-2. Citation.

Response – We consider this to be is a well-known problem with cloud sampling from aircraft.  If the Reviewer has a particular reference, we would be happy to add it.

L286-8. Closure of what concentrations. More explanation.

Response – We stated that is was "closure of number concentrations."

L288. Either values or compares need to lose s.

Response – Removed "values" and changed to compare: line 311.

L289. Why?

Response – The number concentrations of particles > 30 nm compares with the CDNC. That indicates that particles as small as 30 nm participated in the nucleation of the cloud droplets. Even smaller particles may have contributed if 1) one considers the maximum instead of the average and 2) if there are variations in the chemistry that make some of the particles larger than 30 nm insufficient CCN.

291-2. Depends on composition.

Response – We have added that this estimate of S is for ammonium sulphate: lines 314-316.

L294. This comparison seems out of context. Is there a reference for this? If this is so important it should not be just supplementary. These Atlantic measurements should have the same scrutiny as the Arctic measurements.

Response – The Atlantic measurements do have a reference, but this discussion (and supplemental figure) has been removed as it was mistakenly added in response to one of Reviewer 2's initial comments.

L302. There seems to be more variability at all altitudes. Delete the twice. Move overall right after and.

Response – Re-written: lines 317-326.

L327. Delete 2nd the. Add s to equal.
L329. Insert apparently after were.
L332. Delete the twice.
L346-7. Explain this transfer.

Response – If the droplets are formed from moisture evaporated from the polynya and then they settle out over the ice, it effectively is moving water from the polynya to the surrounding ice. We have added "potentially" in advance of "transferring": line 376.

L351. In to within.

Response – Corrected.

L363. Reached low or high?

Response – added "above the surface": line 394.

L368. Delete 2nd the. Delete be.

Response – Done.

L372. Why/how is this characteristic of vertical mixing?

Response – If cloud is generated by adiabatic lifting from below, VMD (and LWC) increase with height and CDNC are constant.

L396. Apparently this is the cloud threshold used in this study. This needs to be stated clearly. This is a rather low threshold, many use 0.03 or 0.1.

Response –Agreed that it is low, but that can be attributed to the nature of the environment, the absence of deep cloud and the sampling of the low cloud, particularly that within the Mauritsen limit. The discussion has been moved to lines 270-273.

L397. Delete first seconds.

Response – Done.

L398. How about a standard deviation.

Response – When are divided between periods and between altitudes, there are insufficient data to produce a normally- or lognormally-distributed distribution. Thus, the sample sizes are insufficient to produce meaningful standard deviations. Instead, we have added the 5[th] and 95[th] percentiles to Tables 1 and 2. The full data ranges are indicated in the figures.

L402. In-situ volume apparently means ambient volume. STP is apparently in parentheses? So state.

Response – We have added that to the Table caption.

L411. For above 200 m is awkward.

Response – Re-written: line 439.

L413. Figs.

Response – Corrected: Line 441.

L414. To above 25 is awkward.

Response – Changed: line 442.

L415-6. Can't this be more precise?

Response – The precise values are in the Table.

L427-8. Needs citation.

Response – The LA clouds (<200 m agl) were close to the surface, and (as stated) some or all may have been technically fogs. They were formed by advection of warmer moist air over a cooler surface (the July 8 LA cloud was likely the result of that process since the cloud moved from Baffin Bay westward along Lancaster Sound where water temperatures were lower), by radiation cooling or by the passage of very cold air over a warm moist surface (also known as sea smoke; the clouds over the polynyas were likely sea smoke with possibly some advection fog component associated with the sea smoke moving from the polynyas over the ice surfaces.) We include this discussion on lines 455-461.

L428-9. Why would vertical motions not be responsible for clouds?

Response – As described above. There will be some vertical motion associated with the sea smoke, but it is relatively small and the cloud (or fog) depths are relatively thin. There also may be some vertical motion associated with orographic features, as we mentioned for one case. Release of latent heat will generate some lifting too, but the LWCs are quite low.

L432. Delete the.
L440. R2 does not match Fig. 7a.

Response – Corrected: line 475.

L444. What is they?

Response – Corrected: line 479.

L445. Delete the thrice. Delete by evaporation of droplets. That phrase seems to sort of contradict without reducing Nc.

Response – Evaporation retained but now preceded by "partial": lines 480-481.

L447. Rate of cooling with altitude? Greater vertical wind (W)? But this will not affect LWC, which depends on distance from cloud base not W.

Response – The sentence is focused on the LA clouds that are impacted more by horizontal transport than vertical transport.

L448-51. Perhaps increased Nc will reduce coalescence but not increased LWC.

Response – If it is significant, collision-coalescence (C-C) will reduce the CDNC. If precipitation results from the C-C, the LWC will also be reduced. Conversely, without C-C, the CDNC and LWC will remain higher.

L451. This is not really an increase but rather higher LWC than otherwise.

Response – Thank you, we have corrected it to read "Changes in collision-coalescence will affect the CDNC and LWC in similar ways: more collision-coalescence, lower CDNC and lower LWC due to precipitation.": lines 485-486.

L453-4. Eliminate commas. Change which to that. This is a restrictive clause rather than a nonrestrictive clause.

Response – Changed, thank you.

L460-1. Explain.

Response – This is just based on the formula for LWC ([4*pi/3][r^3][CDNC}), assuming a density of 1. The LA clouds exhibited an apparent dependence of the LWC on the CDNC, which implies a relatively similar VMD. In Figure 8, we also show that to be true for some of the individual clouds, i.e. it is not just a result of the clustering of all data. The VMD for three of the four cases (7, 8 and 17) are relatively similar, but that for the July 5 case is substantially higher. The VMD is just a means of identifying similarities in the cases.

L461. Relationship plural. Insert is after it.

Response – Corrected: line 496.

L461-3. Why?

Response – As discussed above.

L462. Um to μm.

Response – Corrected: line 497.

L463. Fig. 8 does not have panels.

Response – Corrected: line 498.

L482-3. Of course they do not.

Response – Do you have a particular reference we should include?

L483. Delete necessarily. 2nd to to of.

Response – "necessarily" kept, but "to" changed to "of": line 520.

L488. Less surface area.

Response – condensation sink is correct.

L501-5. This appears to be stated incorrectly, backwards. If not much further explanation is required for these extremely unusual results.

Response – The results are correct. See response to the Reviewer's comment "4" above.

L525-6. How and why?

Response – It is discussed on the lines immediately following the statement.

L536. Insert below 16 after points.

Response – Re-written: lines 563-566.

L537. Last the to these.
L542. Delete the twice.
L543. Comma after clouds.
L544. Insert that before the.

Response – Segment re-written in response to the other Reviewer.

L546. Delete 1st the. Change 2nd the to that.
L544-7. Which differences are referred to here? If it is among the July 5 measurements then ok. But otherwise not.

Response – Re-written: lines 584-588.

L555-6. Has this been demonstrated previously? If so cite. I do not see how you can discount coalescence. Or why this is needed.

Response – With only a few droplets per cc, droplets may grow large enough to fall out without the need for C-C. We are not stating that C-C does not occur, just that it "may" not. This is quite a unique environment.

L558. Must be referring to vertical wind? So state.

Response – As discussed in response to the Reviewer's comment "L427-8", we are not only referring to vertical wind to derive cooling.

L565. Aerosol impact is ubiquitous. Nc below Mauritsen is caused by low NCCN. You cannot get around this.

Response – We agree that overall the aerosol has some impact, and we have noted this on line 602. The question is whether within the Mauritsen limit small increases in the aerosol concentrations increase the LWC, and we see no evidence for that.

L590-1. Again this is the cloud threshold and should be so stated.

Response – added to line 635.

L605. And to with.

Response – added a "the": line 652.

L606. LA clouds with lower Nc should have higher not lower S.

Response – As above.

L609. Insert relatively clean after for.

Response – Added: line 656.

L621-2. All samples exhibit a clear influence of the aerosol. There are differences between the two regimes.

Response – Agreed, we have changed this sentence to read "a clear influence of changing aerosol.": line 669.

L623. Not sure you can call them polluted.

Response – Revised to read "formed in what was likely more polluted air…": line 670.

L625. There is no proof of natural sources.

Response – Smaller particles in the summertime Arctic have been linked to natural sources, and we know have the Willis et al. (ACPD, 2016) reference that discusses a case of aerosol derived from the open waters we were flying over.  We do not explicitly state that these particles are from natural sources, but if such particles are present they have the potential to influence clouds in this type of environment.

Table 1. Explain parentheses. Apparently no distinction of LA and HA?

Response – Values in parentheses are now explained. There is no distinction between LA and HA in Table 1.

Table 2. got to at?

Response – Corrected.

Most figures need tic marks.

Response – Tic marks added to Figures 4 and 5. All other figures have them.

When there are various panels there should not be repetition of every axis label and title. This correction could help allow larger panels. Many panels are small and difficult to read.
Fig. 3. Is a prime example. It is not necessary to write the horizontal axis titles for a-d as they are the same as the panels below. The horizontal axis ranges could also be adjusted so that it is not necessary to write the labels for a-d as they can be the same as those below. B, d,f and h do not need vertical axis titles and labels as they are the same as a, c, e and g respectively.

Response - Figure 3 has been modified accordingly.

Fig. 6 should be landscape with panels side-by-side and then only one vertical axis title and label would be necessary for a. Panels should then be moved closer to each other. Moreover, the figures could be larger.

Response – Done.

Fig. 7 Vertical axis titles and labels for b should be removed and panels moved closer together. Orange appears yellow.

Response – Figure adjusted.

Fig. 8. Is it July 17 or 11? With so few data points probability or significance levels should be shown. This could also be done for Fig. 7.

Response – It is July 17. The caption has been corrected. A discussion of the significance of the slopes at a 95% confidence level has been added to the caption.

Fig. 9. Rescale horizontal for a (0-700) and remove labels. Or make landscape with panels side-by-side with the same vertical scales thus removing b labels. There are no blue data points and yellow seems to be mislabeled.

Response – The x-axes are not the same. Blue changed to black asterisk in the caption.

Fig. 10. Reverse the axes of b so that CCN is the horizontal for both and then the horizontal title and label can be removed from panel a. Or reverse axes of a and plot side-by-side using landscape.

Response – The figures were set up as they are to a) shown the association of CDNC with CCNC and b) the association of CCNC with the aerosol number concentrations.

Fig. 11 caption. In-situ probably means ambient. July 7 appears yellow not orange. July 5 is blue not red.

Response – Adjustments made.

**Reviewer 2**

The topic of this manuscript is very interesting and worthy of study. The authors provide information on clean liquid clouds in the Arctic that will help us more fully understand the Arctic radiative system, particularly during the under-studied, cleaner summertime environments. The study improves our mechanistic understanding of how aerosols affect these clouds and it provides a needed basis for understanding how changes from anthropogenic aerosols affect the Arctic. This is a topic of relevance and importance. In principle, I think this manuscript may be worthy of publication. However, there are a few changes and clarifications that I would like to see before finally recommending acceptance for publication.

First, the paper could really benefit from a more complete discussion of uncertainties and, in some places, a more thorough statistical analysis. I suggest the authors more fully a) substantiate of some of the claims, b) compare to findings from other works, and c) clarify what is known, not known, and what is just a best guess. The section that was most problematic was the discussion of their methods and assumptions regarding calculating activation size from aerosol concentrations and CCN. I provide more detailed comments and suggestions in the specific comments below.

Secondly, the paper would be more useful to the community if the writing and organization were altered to provide more clarity. I was frequently confused throughout the manuscript about what the authors were doing and why, and I had to re-read some parts multiple times over to understand those parts. Some parts of the manuscript could benefit from more selectively discussing relevant information and eliminating irrelevant information; other parts of the paper would be improved by a more detailed explanation of why the reader should be interested. It might be helpful if the authors could consolidate some of the written information into useful figures or tables. There were also, in my opinion, a high level

of references to the supplementary material – to the point that the manuscript does not stand well on it's own. I have given more specific comments and suggestions regarding these points below.

Response – We thank the reviewer for acknowledging the importance of the work and for emphasizing that the summertime Arctic has been under-studied. Also, we greatly appreciate the extremely thorough review. Since the general comments of the reviewer are referred to the specific comments, we respond only to the specific comments.

Specific comments
Most important comments:
1) In the introduction, the authors argue that particles in the 20-100 nm diameter range can become CCN in low aerosol environments. As best I can tell, outside this paper, their initial argument is based only on one previous study (Leaitch et al., 2013). Since a) globally, environments with such low aerosol are fairly rare, b) this hypothesis is not commonly known of or accepted in the community, and c) this hypothesis is a fundamental question/argument to their paper, the authors should describe their reasoning for this argument more fully. For example, if available, the authors should provide more literature references to support the existence of this process. It would also be helpful to expand on why specifically the findings of the Leaitch et al. (2013) paper support the author's hypothesis. Adding this context will make it easier to evaluate the work.

Response – As suggested by the questions set out at the end of the introduction, our work takes more of a phenomenological approach as opposed to setting a hypothesis and testing it; the work is not predicated on Leaitch et al. (2013). We have re-written this segment (lines 70-89), adding references to Hudson (GRL, 2010), Korhonen et al. (JGR, 2010; this reference is used again on line 623), Heintzenberg et al (1994 and 2006) and Lohmann and Leck (2005). We also added reference to Heintzenberg et al. (2015) in connection with the possibility of a cloud-processed minimum in the size distributions. We would appreciate learning about any other relevant references that we have missed.

2) In this paper, the authors make the case that small particles do nucleate cloud droplets based on the following assumptions. First, they assume that the cloud is only incorporating aerosols from below the cloud. Second, they assume that aerosols are preferentially nucleated by size, with larger particles being nucleated first. Thus, they estimate the activation sizes of CCN from the number of CCN in-cloud and the number of aerosols below-cloud in various size classes. However, the various uncertainties in these assumptions need further discussion. What about the impacts of chemistry and hygroscopicity of the individual particles? What about in-cloud aerosol processing or entrainment from above or from the sides? What about precipitation from above? (was there any? Multi-layer clouds are very common in the Arctic). A more clear and straightforward introduction into their assumptions and the uncertainties in these assumptions would be helpful.

Response – We use the aerosol within 50 m of cloud base to define the pre-cloud aerosol for many but not all of the higher-altitude (HA) clouds. Also, it is not entirely an assumption. The purpose of section 2.3.1 (mistakenly labelled as 2.4.1) is to document the reasoning behind our approach to assessing the pre-cloud aerosol for the HA clouds. We see evidence for cloud-top entrainment in some cases (e.g. July 17, Fig. 3d). We don't see a lot evidence that entrainment influenced the CDNC other than through evaporation and mixing with dry air, but there is one case (July 19, Figure 3) where the observations suggest and consequently we assume that the cloud-top aerosol contributed to the CDNC. Carbon monoxide (CO) is used specifically to help trace connections from the various outside regions of the cloud; e.g. Figure 3c and 3e exhibit small increases in CO from inside cloud to above cloud. Further, in

the example of Figure 3c, the CDNC decrease near cloud top, whereas the aerosol concentrations above cloud top are higher. Cases with cloud layers and how we treat them are considered in Section 2.3 and Figure 3: specifically, cases July 19 and 20. The July 20 case is a situation with three vertically stacked cloud layers. With respect to the low-altitude (LA) clouds, we do not use the below-cloud aerosol because the clouds were too low to sample under. In section 2.3.2, we demonstrate our approach to assessing the pre-cloud aerosol for the LA clouds. Further, we discuss in a number of places how our the results differ between using the mean and maximum CDNC, including Section 4 where we state that higher estimates of S will be obtained using the maximum CDNC; that is a consequence of entrainment. We have edited sections 2.3.1 (lines: 298-301; 317-331; lines 341-346) to try to enhance these points.

On the reviewer's second point, we do assume the larger particles are nucleated first and smaller particles subsequently. This paper is only about comparing number concentrations, and variations in the particle chemistry will lead to the possibility that particles smaller that we currently estimate activated. We have added a sentence to the conclusions to this effect (lines 647-650).

The reviewer further questions the absence of some discussion of in-cloud processing, and precipitation. We have added that there was no significant precipitation from the clouds used for study here, while noting that some of the larger droplets in the LA clouds may be considered precipitation; but these are considered in the discussion of data below the Mauritsen limit. In-cloud processing, including chemical transformation, scavenging of interstitial particles and collision-coalescence, reduces particle number concentrations, whereas the process described by Orellana et al. (2011) may enhance particle number concentrations after cloud evaporation.  We now mention in-cloud processing, in the form of chemical transformation, in the introduction as another means of considering activation sizes (lines 84-86). We discuss in-cloud processing in the form of collision-coalescence in connection with the low-altitude clouds, but it is unclear how the other in-cloud processes impact these results, since in estimating the pre-cloud aerosol we theoretically account for effects of previous in-cloud processing. Effects of in-cloud processing on subsequent clouds have no direct bearing on this work; there could be broader implications in terms of climate, but it is beyond our scope. We note that some impacts of clouds on particle number concentrations are one part of another work from this study (Burkart et al., in preparation).

3) Other uncertainties that should be discussed more:

a. Impacts of meteorology on the results. Section 2.3 describes (in way too much detail, I think) the conditions of the study. A lot of information is presented there, but much of it is not directly relevant to evaluation of the key points of the manuscript. At the same time, some very important information is missing, like cloud types (e.g., stratus, cumulus, etc.), cloud sizes, lifetimes, and whether precipitation was occurring. In the summary and conclusions section, it was briefly mentioned that some of the clouds were stratus clouds, but there was no mention of how many. This information should be added sooner and with more detail. There should also be a more thorough discussion of how meteorology and/or surface conditions might or might not affect the results.

Response – As above, defining the 'pre-cloud' aerosol is a critical aspect of these sorts of studies, which is the motivation for section 2.3.  We believe that it is directly relevant, adding to the integrity of the paper.

We agree that an earlier discussion of the cloud types is important, and we have added that in the preamble to section 2.3 (lines 252-259).  The higher-altitude clouds were either stratus or stratocumulus. The low-level clouds were fog or stratus.

We apologize for not specifying that the clouds were absent of precipitation as indicated by the 2D imaging probes. We have included in lines 252-259 a statement to indicate this.  An addendum to

this is that droplets in a couple of the low-altitude clouds were very low in number and relatively large in size (30-40 μm). Based on the settling speeds of such droplets, some might view them as precipitation.

General aspects of the meteorology are discussed in section 2.3, and one clear effect of the meteorology is the cleansing of the air. That effect is clearly an important aspect of these observations and it is discussed elsewhere in the paper. We enhanced discussion about the nature of the LA clouds on lines 455-461. Otherwise, the paper is about the effects of particles on the clouds, and in that regard we think we have sufficiently addressed the central impacts of the meteorology.

b. Reliability of in-cloud measurements. Were the in-cloud CCN values reliable? What about the aerosol measurements (I know the authors said these were not used in the analysis, but they are shown in Fig. 3)? From what I understand, lot of instruments do not function as well in cloud as out of cloud.

Response - We had stated that the in-cloud aerosol measurements are shown for completeness only, but we should have made that statement earlier. The discussion of the in-cloud measurements is now on lines 276-286. From inside of cloud, only the CDNC, CO and thermodynamic measurements are used. It would be nice to be able to say that the aerosol (including CCN) measurements inside cloud are reliable in order in order to help with number closure, but we can't. Unless a sampling system is employed that is designed specifically to enable in-cloud sampling, which was not the case here, in-cloud measurements are unreliable due to issues of drying and partial drying associated with the inlet and a particular instrument as well as droplet shattering on the inlet.

c. The ability to adequately characterize the influence of smoke from the CO and aerosol data alone (e.g., P.27, l. 595). On P. 14, l. 311 Koellner et al., 2015 is referenced to say that the very slightly elevated CO was from BB. However, I do not think it is appropriate to cite a non-peer reviewed conference abstract to make a point here. Also, CO levels of 100+ ppbv are still quite low compared to other measured CO values in the Arctic, and other summertime studies still count values of CO above this level as "background air" (e.g., Sakamoto et al. (2015, doi:10.5194/acp-15-1633-2015)). Also, particle concentrations were still quite low, weren't they? The authors might consider restating this sentence to mention that the CO levels were elevated compared to your other days, perhaps due to dilute BB, but that they were still pretty clean. From what I understand, photochemical destruction of CO would lower average CO values in the summer compared to other seasons, making it additionally difficult to compare aerosol:CO values to assess influence from biomass burning smoke. Also, there are other source of CO and aerosols besides smoke, and being a gas, CO is an imperfect smoke aerosol tracer to begin with. Anyways, I think the claim that these air masses were influenced by smoke is more uncertain than is currently presented.

Response – Thank you. We have moved the Koellner et al. reference to inside the text so that it is not considered a peer-reviewed reference (lines 335-339). The basis for assuming contributions from biomass burning are 1) the trajectory transport pattern shows the air arrived from the location of forest fires in the NWT of Canada, 2) the CO does increase and 3) the size distribution to larger sizes. These points suggest a BB contribution. We feel that a detailed discussion of the CO is for another paper.

d. P. 24, l. 536: there is an awful lot of scatter (understandably) in figure 11; thus I think there should be some discussion of the uncertainties of the 16 cm-3 limit the authors use as their estimate of the Mauritsen limit. Relatedly, on this line I would not call it "the Mauritsen limit", but rather, "our best-guess Mauritsen limit."

Response – We changed "the Mauritsen limit" to "our best estimate of the Mauritsen limit" on lines 559-561 and 564. The consequence of using a value of 10 cm$^{-3}$ is discussed: lines 592-595.

4) Other key information that was missing or unclear:
a. In the introduction/methods there was not a clear, concise description of how the questions would be tested, and how the methods would provide useful information. Please add.

Response – We have modified the general approach to addressing the questions: lines 126-129.

b. Given that 62 different clouds were examined, it was not clear to me why a select few were chosen for more detailed analysis. Please provide this information.

Response – There were 62 cloud "samples". The numbers of individual clouds were fewer, since more than one profile was conducted through a cloud layer. We have added a few words to clarify this and the reason for the selections on lines 270-278.

c. Why were LA and HA clouds specifically separated at 200 m?

Response – We apologize that this was not made clear, and a discussion of this point is now found on lines 278-281. The LA clouds were in the boundary layer, indistinguishable from the surface in flight, and we were unable to sample below the cloud due to proximity to the surface. An additional reason for the separation is that we were able to sample below all HA clouds. We have added further discussion of the nature of the LA clouds to section 3.2 (lines 455-461).

d. p. 13, l. 292: "for 20 nm particles to activate, the water supersaturations in the bases of the clouds would have had to reach about 1.5%." How was this calculated?

Response – This is an estimate based on Köhler equilibrium theory. We now indicate this on lines 314-316.

5) There is a heavy reliance on the supplementary figures, and it makes it harder on the reader to have to download another document and scroll through just to understand the main text. It would be better to just put the relevant figures in the main text.

Response – We moved some figures to the supplement as a result of previous review comments. We prefer to leave the paper as is, but would be happy to move a figure the Reviewer considers particularly relevant.

6) I think the authors are actually underselling a couple things:
a. the fact that no one has observed this Mauritsen hypothesis in the field before. This point makes their work more exciting and I think they could emphasize it more.
b. the fact that the clouds they are describing are so clean. As noted in the general comments, a better baseline of clean clouds provides a much-needed basis for understanding how changes from anthropogenic aerosols affect the Arctic and this lends importance to the study. This point could also be emphasized more.

Response – Thank you. We have re-written the abstract to try to highlight these points a little.

Other specific comments:
7) Statements that should be backed up more with further statistical analysis:
a. p. 14, l. 304: "The VMD exceed 20 um compared with a peak value of about 15 um for the July 7 profile." Is this statistically significant difference? Also, please give more information on why this statement is relevant to the paper.

Response – The question as to whether the difference in the VMD is statistically significant is interesting. Because of the similarities in the LWC profiles and the inherent relationship among CDNC, LWC and VMD, we can address the question using both the CDNC and the VMD. The CDNC range between 150 $cm^{-3}$ and 250 $cm^{-3}$ in the case with the lower VMD and 20 $cm^{-3}$ and 75 $cm^{-3}$ in the case with the higher VMD. Since there is no overlap of the CDNC, the means are significantly different. We can also look at the range of the VMD for each case. In one case, the VMD range from about 7.5 um to 15 um for the LWC varying from 0.05 to 0.25 g $m^{-3}$. In the other case, they range from about 10 um to 20 um for the same range of LWC values. Again, there is no overlap of the VMD for similar values of the LWC indicating that the VMD are significantly different. It is relevant only because it is a simple demonstration of the basis for the cloud albedo or Twomey effect. We have re-written this discussion accordingly: lines 317-331.

b. P.22, l. 491: where did you get the 40% value from? Is this statistically relevant?

Response – This is based on the median of the 38 samples of HA cloud (Table 2) and the regression in Figure 9b. We are unclear what hypothesis we should be testing for statistical significance.

c. P.23 l. 503: you say there is a significant difference. What methods did you use to determine the significance, and how significant is it?

Response – In Figure 10a, 14 of the 16 points lie well below a 1:1 line. Therefore, the mean supersaturation inferred from the LA points is less than 0.6%.

8) Other missing or unclear information:
a. P.24, l. 524: "the vertical dashed green line represents the Mauritsen limit below which the cloud may produce a net warming for an increase in the CDNC. " 1) in the earlier text, this limit was estimated as 10 cm-3, and in the figure it is estimated at 16 cm-3. Since you bring up the figure here first, can you please describe here why there is this difference, and not on line 535, where it is mentioned first? 2) Also, please state why you say in the above text that below this line the cloud can produce net warming. Are you relying on the Mauritsen model results for this statement? If so, please say so.

Response – Thank you. We have addressed both concerns on lines 559-566.

b. P25, l.558-560: "Variations in other processes, such as mixing or the rate of cooling, may be responsible for the correlation of CDNC and LWC. In these cases, there is sufficient aerosol so as not to be a limiting factor for the CDNC." How do you know that?

Response – Thank you. "sufficient aerosol" is incorrect. In response to this and a comment from Reviewer 1, we have replaced that statement with "It can be argued that some aerosol must exist for these clouds to form, but these observations show no association of changes in either the CDNC or LWC with changes in the aerosol": lines 602-604.

c. P.26L l. 577: "The lower- and higher-CO points overlap over a CDNC range of 16 cm-3 to 160 cm-3." The way this is phrased is a little confusing. I am not sure how it tells you "that 20-100 nm particles from natural sources can have a broad impact on the range of CDNC in clean environments." (l. 579-580)

Response – We have changed it to read: "In this clean environment, the contributions from 20-100 nm particles have a broad impact on the range of CDNC, affirming the large uncertainty associated with estimating a baseline for the cloud albedo effect discussed by Carslaw et al. (2013)." Lines 623-626.

d. P, 4, l. 81, "remote sensing particle Na do not reflect the size distribution" can you reword to make this sentence clearer, e.g., small particles are not observable, or something like that? I think this could be a very important point for your argument, so it is important to make sure it is clear. (also, please add a reference)

Response – In response to this comment and others, we have re-written this segment: lines 76-86.

e. P. 11, l. 236: can you explain what you mean by saying that clouds were sampled more frequently during period 1 because of "better visual contrast"? I am confused.

Response – The issue is of course that ice and snow as backgrounds can make visual definitions of cloud more difficult, and this is a particular problem for the Arctic. During part 1, the clouds, in particular the low-level clouds, were more clearly defined compared with part 2 when high overcast in the region reduced contrast. We have re-written the statement as "Cloud was sampled on eight of 11 flights, more frequently during period 1 because of overall better visual contrast between clouds and surrounding surfaces as well as because period 2…" Lines 236-237.

f. Lines 292-300: It is not really explained why the authors are comparing and contrasting the July 7 case with some other unrelated cloud case. Can this be made more clear, or taken out?

Response – This was added in response to the Reviewer's preliminary comments for ACPD that asked for comparisons. Clearly, we misunderstood. We have removed it from the main text and supplement.

9) Abstract: The authors might consider adding a sentence on what this all means for the bigger picture.

Response – We have revised the end of the abstract in response to your comment 6 above. Otherwise, we feel that it is too easy to speculate incorrectly.

10) I agree with Dr. Hudson that the authors should consider using a different to describe cloud droplet number concentration and concentration for cloud condensation nuclei concentration. Currently, the authors use CDNC and CCNC, respectively, and these abbreviations look quite similar, which could lead to some confusion.

Response – As in our response to Reviewer 1, we respectively decline this suggestion. To reduce the potential for confusion, we now write CCNC as CCNC(0.6%) where we refer to our measurements.

11) p. 13, l. 288: "The N30 values compares most closely with the mean CDNC…" [I did not see that in Fig. 3a?]

Response – The N30-100 must be added to the N100 to derive the N30. It was discussed early in a general sense, but we have added it explicitly to line 311.

12) p. 11, l. 246, why are all Na concentrations for particles < 100 nm diameter taken from the SMS, when you also have TSI data down to 5 nm from a different instrument?

Response – Total number concentrations of particles larger than 5 nm were obtained from the TSI 3787, but that is without any information on size distribution. The size distribution information from 20 nm to 100 nm was obtained from the BMI SMS, and size distributions from 70 nm to 1 μm were from the UHSAS. Overall, the different measurements compared reasonably well. See our response to your next comment.

13) Did SMS and UHSAS match up fairly well in their overlapping section and generally speaking? Because each might have different artifacts, do you have evidence that these artifacts are either small, or at least similar?

Response – Calibration information is given in the supplement. Also in the supplement we show an example of the comparison between the UHSAS and SMS for one flight (Figure S3). For the nominal overlap region of the two instruments (70-100 nm), we had to use the 77-100 nm range from the SMS to derive the closest agreement with the UHSAS over 70-100 nm. We use the SMS up to 100 nm because it provided slightly higher number concentrations over the 70-100 nm range, making our results for the activation sizes more conservative.

14) p. 12 l. 272: you say it is one of several profiles through the same cloud. How do the other profiles compare?

Response – They compare well. They are represented by most of the points clustered between CDNC values of 150 cm-3 and 200 cm-3 in Figure 9b. We have re-written it to say "One of several similar profiles…" on line 294.

15) P. 16, l. 351: "The CO mixing ratio is slightly higher in the cloud (81 ppbv) than above (79 ppbv)." I doubt that is statistically significant and outside the error of the instrument. I suggest taking this sentence out.

Response – We have left it in to reference the CO measurement, but we have added that the difference may not be significant: lines 381-383.

16) Fig. 8: Are each the points the mean (or median, or something like that) of an individual cloud? Can that be stated more clearly in the caption? Why are clouds within each flight considered separately from each other?

Response – The points are taken from the 62 points that are the basis for the analysis. For clarity, we have added the following to the beginning of section 3: "All the analyses are based on the 62 cloud samples discussed in section 2.3. The LA cloud subset is comprised of 24 samples and the HA cloud subset consists of 38 samples." See lines 421-423. A selection of the LA clouds are considered individually in order to demonstrate that the linearity between the LWC and CDNC occurs within some individual clouds, and that it is not just a result of combining the LA points.

17) p. 21, l. 463: There is no figure 8a. Did you mean Fig. 8? If so, can you be more clear about why LWC and VMD indicate anything for Fig. 8?

Response – Sorry, it should read just Figure 8, and has been corrected on line 498. The LA clouds exhibited an apparent dependence of the LWC on the CDNC, which implies relatively similar VMD. In Figure 8, we wish to demonstrate that is true for the individual clouds and not just the clustering of all data. The VMD for three of the four cases (7, 8 and 17) are relatively similar, but that for the July 5 case is substantially higher. The VMD is just a means of identifying similarities in the cases.

Technical remarks
1) Fig. 1: can you add an inset that shows Canada, and where Resolute Bay is relative to the larger region? It would also be helpful to show which areas are land vs. ocean.

Response – Added.

2) It would be useful to edit Section 3.4.1 for clarity (it is currently 13-sentence paragraph, so breaking it up would help).

Response – The paragraph is now separated at line 596.

3) Section 2.4.1 could be substantially shortened, and made into a table or figure to better elucidate the main points the author wishes to get across. One suggestion would be to add this information on supposed activation sizes into Fig. 3.

Response – Section 2.4.1 was mistakenly numbered in the original manuscript. It should be 2.3.1, and has been corrected in the revision. We agree that this section is difficult, but it is the basis for the analysis and therefore we believe it is important. The individual cases of the section are relatively short. While the discussion in section 2.3.1 focusses on how we estimated the pre-cloud aerosol size distribution, it is not the point of the paper to document precise activation sizes, since that level of detail has meaning only in the specific circumstances. Overall, we simply wish to show that when considering cloud droplet nucleation in the summertime Arctic, one must take into account the possibility that particles as small as 20 nm may be a source of viable CCN. As above, we have added to section 2.3 the reasons for this section: lines 276-278.

4) p. 11, l. 242: "Number concentrations [of particles] greater than 100 nm"?

Response – Added on lines 243-245.

5) p. 13, l. 289: by, "in the mean " do you mean "on average"?

Response – Changed on line 312 and 346.

6) p. 14, l. 302: more variability in what specifically?

Response – This segment has been re-written: lines 317-326.

7) Fig. 4a: is CO the same axis as altitude, as in Fig. 4b?

Response – Thank you. The CO label has been added.

8) Fig. S7: Caption says LWC and VMD should be shown, but I only see CDNC

Response – There are three vertical profile plots side-by-side: one for CDNC, one for LWC and one for VMD.

9) Fig. 9: caption says blue, but the points are yellow and black symbols.

Response – Thank you. That has been corrected.

10) P22, l48: "The CDNC are plotted against the N50 in Fig. 9b showing that the mean activation size [of HA clouds] was often close to 50 !nm."

Response – Thank you. "of HA clouds" has been added to line 526.

11) Figure 10: what is this? All the samples? Caption and discussion in the text need to be more clear.

Response – Section 3 begins with indicating that 62 cloud samples were defined. All the points in the plots are based on that data set. We mention on line 435-436 that the CCN data set was limited to 44 points due to instrument problems. We have explained this again in the caption of Figure 10.

12) Figure 11: you refer to red dots, but I think you meant blue dots

Response – Thank you. That has been corrected.

13) Figure S8: do these points represent averages of single clouds? Please specify.

Response – These points are the subset of the 62 that fall below the estimated Mauritsen limit. That has been added to the caption.

14) P.25, l. 554: "In this low CDNC environment, where cloud droplets may grow large enough to be gravitationally removed from the cloud without the support of collision-coalescence." Reference?

Response – We believe that we are referring to something that is now common knowledge: for a given LWC, if the CDNC are low the droplets sizes will be high. 40 μm droplets will settle out if conditions are right (i.e. close to the surface and low turbulence). Also, it is stated as a possibility derived from these observations, not a definitive result. Collision-coalescence is also discussed in this context on lines 485-486.

15) P. 25, l.: The reader is referred to the Fig. 11 caption. This information would be more appropriate for a table or for the text.

Response – We have moved that information to the text and re-written the discussion on lines 584-588.

16) P.26, l. 577: The reference of Burkart et al., 2015 is another conference abstract and is not a suitable reference.

Response – On line 621, we have replaced that reference with Willis et al. (2016) that is currently in ACPD.

18) Table 1 and 2 can be combined into 1 table.

Response - There are some details in the "Measurement" column that differ between the two tables. Also, combining them to produce 8 columns of means and medians can make it more difficult to sort through the numbers, especially since we added percentiles based on a request of Reviewer 1.

19) P21, l. 475: It is stated that, "As shown in the plot of CDNC versus N100 (Fig. 9a), particles smaller than 100 nm activated in most cases", but on p.11 it is said that, "Number concentrations greater than 100 nm (N100) are taken from the UHSAS." So on line 475, instead of "smaller than", did you mean "greater than"?

Response – We intend to say that particles smaller than 100 nm are activated because the CDNC are larger than the N100. We have re-written it (lines 510-512) to read "The CDNC are plotted versus N100 in Fig. 9a, separated between LA and HA samples. The CDNC are most often higher than the N100 and more so for the HA samples, which indicates that particles smaller than 100 nm activated in most cases and most often in the HA clouds." Hopefully that is clearer.

20) P. 27, l. 612: I'd suggest saying ~16 instead of simply 16.

Response – We refer to it as an estimate.

---

## Author Comment (AC2) · 10 May 2016

The comment was uploaded in the form of a supplement:
http://www.atmos-chem-phys-discuss.net/acp-2015-999/acp-2015-999-AC2-supplement.pdf

---

## Referee Report (RR1)

The authors have satisfactorily answered my comments.  As will be shown I have some problems with interpretations or explanations of Figs. 3 and 4.  Nonetheless, this manuscript should definitely be published.  Field results of such important topics as these need to be made available.

I did detailed editing for the first half of the manuscript.  This was intended and resulted mostly in text reductions.  In almost every instance maximum numbers of words were used.  This was not just the definite article as mentioned in my first review but also prepositions such as of and in.  However, it occurred to me that perhaps for some reasons these authors do not want to reduce the length of the manuscript.  Thus, since my editing might be in vain I stopped most editing half way through the manuscript.  If this notion is incorrect the authors can follow my examples of the first half to edit the 2nd half accordingly.

L32.  Delete the.
L38.  Delete the.
L39.  And to with.  Change are used to infer to imply.
L40.  Delete respectively.  The two sets are in order.
L43.  Delete the twice.
L44. Delete the.  Delete in the.  Change changes to variations.  Move aerosol in front of variations.
L45.  In to within.
L46.  From to between.  To to and.
L47.  Delete the.
L52-3.  Change and lower during to than in.
L53.  Delete of.  Particles singular.  Move transport after particle.
L54.  Delete the.  Delete 2nd of.  Move focus after chemistry.
L55.  Delete in the.  Move Arctic in front of research.  Delete the.  Add time to spring.  Delete period.  Delete from.  Move transition after summer.
L56.  Delete 1st 2 thes.  Opportunity plural.  Delete in.  move changes after chemistry.
L57.  Delete last the.
L59.  Delete of the.  Move Arctic on front of warming.
L72.  Change aerosols to particles.
L73.  Delete sources.  Delete for.  Move of after nuclei.
L74.  Drop s of towards.
L76.  Delete much.  Change sometimes to often.
L82.  Explain surface-active.  Does this mean hygroscopic?
L83.  Delete 1st the.
L84-85.  Move summertime and Arctic in front of environment.  Delete of the.
L85.  Particles singular and move in front of concentrations.  Delete of.  Above to larger than.
L87.  Delete nm.
L89.  Delete much.
L91.  Period after parentheses.  And to Moreover.
L92.  At to to.  ing to e.
L96.  Delete made.

L97. Delete that.

L98. Delete the. Delete in the. Move CDNC after model.

L99. Delete will. Above to for.

L100. Insert greater than before 10. Delete in. move increases after CCN. Delete of the. Change phere to pheric. Move cooling after atmospheric. Change the to This.

L101. Move threshold after concentration.

L101-2. Change and it is noted that the value of 10 cm$^{-3}$ to although this. Insert a after not. Insert limit after universal.

L103. Delete 2$^{nd}$ the. Add time to summer and move in front of microphysics. Clouds singular and move in front of microphysics.

L107. Insert Yum and Hudson, 2001 and Wylie and Hudson, 2002.

L108. Delete 1$^{st}$ the. Add time to spring.

L109. Aerosols to particles.

L111. Delete of. Aerosols singular and move in front of observations.

L112. Delete during. Add time to summer and move in front of Arctic.

L113. Delete of. Aerosols singular and move in front of measurements. Add time to summer. Move Arctic in front of clouds.

L114. Period after forcing. Insert They to begin next sentence. ing to ed.

L115. Delete about. Move forcing from these plumes after maximum. Half is approximate.

L117. Among to between. Insert and after coupled. Move uncoupled in front of to. delete versus those. Period after surface. Change but to They. Delete of the. Move observations after microphysics. Insert their after and.

L119. Characterization plural. Was to were.

L119-20. Move June in front of low. Delete in.

L121. Delete the. Delete in the. Tops singular. Move cloud top in front of CDNC.

L121-2. Remove quotes. Move aerosols to end of sentence.

L122. Delete the twice. Bases singular. Insert cloud in front of base.

L123. Delete does. Add s to influence. Delete the.

L124. Delete the twice. Delete in. add time to summer. Cloud plural and move after Arctic.

L127. Aerosols and clouds singular and move in front of observations. Remove of.

L128. The to this.

L153. Delete are.

L157. The to these.

L162. The to these.

L165. Add s to detect. Move detects particles right after that. Uses to using. Delete of. Move scattering after light. Delete to.

L167. Delete a reduced pressure of.

L168. Last The to This.

L169. Delete of. Move measurement in front of stability.

L170. Delete the. Delete of. Particles singular. Move hygroscopicity to end of sentence.

L171. Using to with.

L173. Period after parentheses. Delete and.

L176. Dimensions to dimensional. Move two dimensional in front of Cloud. Delete in. sized from about to between. To to and.

L177. Using to with. Delete For. Delete present. The to this.

L177-8. Move this study to the end of this sentence. insert from after phase.

L179.  Period after parentheses.  Delete and.
L182.  Delete The.
L183.  Use CO.
L184.  Delete at.  Move 150nm in front of excitation.  The to This.
L191. Measured to done.
L192.  The to This.
L198.  Semicolon to period.  Insert However before the.
L199.  Insert or below after in.
L200.  Are to was.
L201.  Are to was.  Is to was.
L205.  Delete of.
L206.  Delete the.
L207.  Delete of.  Particles singular.  Move transmission after particle.  Approximately to near.
L211.  Delete at.
L211-2.  Move exhaust tube in front of flow.
L212.  Move flow after tube.  Delete the four times.  Delete of twice.  Delete at.  Delete allowed.  Delete last flow.  Delete ly.
L213.  Move flow after intake.  Delete at the.  Period after TAS.  Delete and.
L216.  Delete 1$^{st}$ the.  Delete of the.  Move aircraft in front of forward.
L217.  Change lowered to reduced.  1$^{st}$ the to this.
L220.  Delete to.  Move Analysis in front of Approach.
L223.  Beginning to between. Delete ending July.  The to These.
L224.  Delete relatively.  Delete the.  Relatively and distinct are opposites.
L225.  Delete The.
L226.  Change calm to light.  Change varying to variable.  Insert the before south.  Delete to north.
L233.  Period after parentheses.  Then insert This was.  Delete in part.  Possibly is enough of a hedge.
L237.  The to this.
L239.  Delete of.  Flying to legs.
L240.  Above to altitude.
L241.  Delete the surface.  Surface here must me sea level.
L245.  Delete surrounding.  Period after surfaces.  Change as well as to Furthermore.  Delete because.  Insert was after 2.  Insert by after marked.
L246.  Put Fig. 2 in front of panel.  Delete in.
L247.  Flight plural.  Delete plans were.  Change towards sampling to on.
L251.  Change greater to larger.  Delete for.
L252.  Delete The.
L253.  Over to between.  Dash to and.  Delete data, which are.
L253-4.  Move Fig. S3 in front of example.
L254.  Change shown in to and.  Particle singular and move in front of number.  Delete of.  Delete 1$^{st}$ nm.
L258.  In to within.  Move study in front of area.  Delete of.  Move Within the study area to beginning of sentence.  Delete when they.  Change ideally to mostly.  Ascending to ascents.
L259.  Or to and.  Descending to descents.  Delete through them.  Delete the.  Base plural.  Delete of.  Clouds singular.  Move bases after cloud.

L260-1.  Move only liquid phase clouds after µm.

L262.  With the caveat to except.

L264.  May to might.

L265.  Period to whereas.

L266.  Move July 7 in front of stratocumulus.  Delete sampled on.  But to though.

L270.  Insert and when after where.  Change was clear and achievable to could be observed.  Clear is a poor word choice to describe cloud base.  Change semicolon to period.

L272.  In to within.  Comma to and.

L272-3.  Move in flight in front of indistinguishable.

L273.  Period after parentheses.  And to Thus,.  The to such.  Cloud plural.

L278.  Are to were.

L281-2 and elsewhere.  Points is not the best word choice unless you are referring to elements of a figure.  Sections or segments are alternatives.  Or just refer to data without another word.

L282.  Is to was. The to These.

L288.  Delete the.

L289.  Move valid in front of in-cloud.  Delete considered.  Delete inside of cloud.

L290.  Thermodynamic plural.  Delete measurements, it is used twice in this sentence.  Delete the.  In to within.

L293.  On to upon.  Insert Hudson and Frisbie [1991] and Hallett and Christensen [1984].

L303.  Delete on July 7 sampled.  Delete 1st the.

L304.  Change and the to while.

L305.  Period after altitude.  Insert These before features.  Change common to the to characterize.  Move formation after cloud.  Delete of.  Change and indicating to so.

L306.  Delete cloud.  Delete in air rising.  Insert below after from.

L308.  Delete 1st the.

L309.  Delete the.  In to Within.

L313.  Delete 3rd the.  Insert Hudson [1993] at end of sentence.  Change The to Thus.

L315.  Delete 2nd the.

L317.  Insert corresponding before N5.

L318.  Delete the.  Move closure after concentration (singular).  Delete of.

L320.  Change down to about to as small as.

L320-1.  Move based on maximum CDNC to beginning of sentence.

L321.  Delete the.  Delete of.  Move particles after sulphate.

L322.  Change the to that.  Delete of the.  Clouds singular and move in front of bases.

L324.  At to of.

L325.  And to while.

L326.  Change but there are to except that.  Insert is before more.  Insert broken after more.

L326-7.  Move the July 17 profile right after except that.

L327.  Delete what is left of this line.

L328.  Delete adiabatic lifting.  Insert lower LWC before intervals.  Delete with decreasing LWC.  Change associated with to due to.

L329.  Change the to cloud.  Delete of the stratocumulus.  Profile plural.  Last the to this.

L330.  Move LWC in front of peak.  Delete in the.  Change below to from.

L331.  Move CO in front of increase.  Delete in.  insert at before about.

L332.  Move the in front of erosion.  Change was to went.  In to into.  Change case to cloud or clouds.

L333. What aerosol increase above cloud? Aerosol decreases at many levels. Delete the.
L334. This is true at a greater altitude range.
L335. This is not shown in the figure. Delete last the.
L336. Larger to higher. Of to between.
L340. 1$^{st}$ the to a.
L342. Last the to this.
L343. Period after study. Delete and. Insert that before BB.
L347. Add ed to reach.
L348. Period after layer. Insert This is.
L349. Reduced to lower. Increased to higher.
L350. Semicolon to comma. Insert which is. Increase to higher. Delete in. insert concentrations above than after aerosol. Delete between. The to this. Delete and.
L351. Delete above the layer. N50 is apparently not shown in the graph! I do not see these numbers.
L354. Delete sized. Delete 1$^{st}$ nm. Insert diameter after nm.
L355. Insert comma which after CDNC. Down to about to as small as.
L356. Reduced to lower.
L357. Insert below cloud after of. And to with. Above cloud CCN does not show this.
L358. Delete case of a. delete in.
L358-9. Move LWC in front of variations.
L359. Delete the. Suggests to suggest. Looks higher than 49.
L361. The to These.
L363. I do not see this in the figure.
L365. Pre to below.
L365-7. It is problematic to get valid aerosol measurements of any kind in the narrow layer between these clouds.
L366. 44?
L367. 52?, seems higher than 34. Seem higher than 66. Seems higher than 35.
L369. For the lower cloud layer yes, but not so sure about the other two.
L372. Delete 2$^{nd}$ July.
L373. Delete and July.
L377. Looks much higher in Fig. 4a.
L378. Apparently below cloud is to the right? 16:45 to 17:09 and beyond? This needs to be stated. N100 appears to be 2. 0.6% ammonium sulfate is 40 nm diameter. 100 µm ammonium sulfate is 0.1%.
L385. Explain these 7 samples. This is not obvious from the figure.
L389. Where is N100 this low? To the right and left sides of Fig. 4b N100 is 50 or more! Lower values are seen within cloud. All measurements where altitude is shown seem to be within cloud and thus invalid for aerosol measurements.
L390-1. Delete due to instrument problems.
L391-2. These are not the numbers that appear on Fig. 4b.
L400. Delete from. East to easterly. Delete to west.
L404. The to This.
L414. The to These.
L418. The to these.
L427. The to these.

L467.  The to these.
L533.  Insert S after %.
L572.  Reach above to exceed.
L578.  Insert that before 1st the.

Hallett, J., and L. Christensen, The splashing and penetration of raindrops into water, J. Rech. Atmos. 18, 226-242, 1984.

Hudson, J.G., 1993:  Cloud condensation nuclei near marine cumulus. *J. of Geophys. Res.,* **98,** 2693-2702.

Hudson, J.G. and P.R. Frisbie, 1991:  Cloud condensation nuclei near marine stratus*. J. of Geophys. Res., **96**,* D11, 20,795-20,808.

Wylie, D., and J.G. Hudson, 2002:  Effects of long range transport and clouds on cloud condensation nuclei in the Springtime Arctic. *J. Geophys. Res.*, **107(D16), 4318,** doi:10.1029/2001JD000759

Yum, S.S., and J.G. Hudson, 2001:  Vertical distributions of cloud condensation nuclei spectra over the springtime Arctic Ocean.  *J. Geophys. Res.,* **106,** 15045-15052**.**

---

## Author Response (AR2)

Responses to Reviewer Comments

General – We are grateful to the reviewers for their interest and helpful critiques. We appreciate their time and effort in helping to improve this manuscript. Responses to the review comments are highlighted in yellow.

**Reviewer 1**

Major Comment from Reviewer 1
Overall, I am very pleased with how thoroughly the authors addressed my first comments, and I am ready to support the paper for publication, as long as one major comment is addressed. As an aside, I congratulate the authors in that I have overheard sincere community interest in the ACPD paper from my colleagues.

Major comment based on Specific comment 4b: Given that 62 different clouds were examined, it was not clear to me why a select few were chosen for more detailed analysis. Please provide this information.

Response – There were 62 cloud "samples". The numbers of individual clouds were fewer, since more than one profile was conducted through a cloud layer. We have added a few words to clarify this and the reason for the selections on lines 281-290.

New comment from the reviewer: Ah, I clearly misunderstood that there were 62 independent, separate clouds sampled (to me, the terms "cloud averaged data point" or "Forty-five cloud samples" (e.g., line 650) imply independent samples, taken from separate clouds). Now that I know that the points are not averages from single clouds, but rather averages of penetrations within clouds, I suggest replacing the terminology of "cloud-averaged data points" and "cloud samples" in the paper. Despite the new text added on lines 281-290, these terms may mislead other readers (particularly, but not only, if they are skimming the abstract and conclusions), and there are more accurate terms that can be used. At minimum, I suggest rephrasing to something like "the average of individual cloud penetrations".

However, I strongly believe more analysis may be required here. Individual cloud penetrations within the same cloud (e.g., Fig. 5), should not be treated as independent samples, as they are in the text. Failing to do so will bias the results (e.g., Tables 1 and 2) toward the characteristics of clouds with the most profiles sampled.

As it was not stated how many independent clouds were sampled, particularly for the HA cases, it is hard to fully evaluate the impact of this issue on the results. It would be helpful if the authors could provide as much information as they can about how many independent clouds they actually sampled. Something along the lines of what the authors show with the new supplemental Figure 7 is helpful, but alone it is not sufficient. I also strongly recommend that the authors redo the statistics with independent clouds separately. If the findings are not still the same, further discussion in the text will be needed.

Author response - We have further clarified our approach using the phrasing "averages of individual cloud penetrations", as suggested by the reviewer. We did consider that our approach biased the statistics in the manner the reviewer discusses, but we believe our approach is reasonable in terms of the effects of aerosols on clouds for climate. Cloud coverage is a major factor in terms of cloud influence on climate. If we generate one point from one large cloud and plot it with one point from one small cloud, the "climate-impact" of a regression is biased inappropriately to the small cloud. It is difficult to conduct these measurements in an ideal manner. Our approach was to sample every liquid cloud that was available to us as much as possible, and it was random in so much as we sampled cloud as the opportunity presented itself along our flight path without purposely seeking cloud. Although we are not saying this approach is perfect, in some way it takes into account clouds that are more extensive than others, and that bias we feel is appropriate from a climate perspective. We have added some discussion to this effect on lines 266-271 of the revised manuscript.

Minor comments from Reviewer 1

• Regarding the following newly added sentence: "Variations in particle chemistry will induce some variance in these results, but because activation diameters are estimated starting with larger particles and moving to smaller sizes, changes in chemistry only offer the possibility of activation of particles still smaller than estimated here." My understanding is that size and hygroscopicity are two opposing constraints on activation, and thus depending on the situation, activation could depend on either. I suggest the authors rephrase this sentence.

Author response – We are not of the opinion that size and hygroscopicity are "opposing constraints", since they tend to act in the same way; nor do we feel that treating them as independent (which some people seem to do) is appropriate. Regardless, our comment is simply about number conservation. Given the nature of the particle number distribution, if we start from the largest size and move towards smaller sizes to estimate an activation diameter, the true activation diameter can only be larger than our estimate if there is a significant error in our number measurements; hopefully, we have demonstrated otherwise. Chemistry can only dictate that particles smaller than our estimate activated. That of course would happen at the expense of some larger particles, and we have added that point of clarification to the sentence you mention.

• Why add in the Koellner reference at all? It is fairly long and not as descriptive as simply saying, as the authors did in their response, that: 1) back trajectories suggested that the air arrived from Canadian forest fires, 2) there was an [~20 ppbv?] increase in CO concentrations compared to other days sampled that month, and 3) the size distribution transitioned to larger sizes, which suggests some BB influence?

Author response – Thank you for the comment, but it is our preference not to make this change.

• Line 83: "Lohmann and Leck (2005) hypothesized the need for highly surface- active particles to explain CCN [activity?] at S less than 0.3%."

Author response – Corrected.

**Reviewer 2 (Dr. Hudson)**

Minor comments from Reviewer 2 (Dr. Hudson)

The authors have satisfactorily answered my comments. As will be shown I have some problems with interpretations or explanations of Figs. 3 and 4. Nonetheless, this manuscript should definitely be published. Field results of such important topics as these need to be made available.
        I did detailed editing for the first half of the manuscript. This was intended and resulted mostly in text reductions. In almost every instance maximum numbers of words were used. This was not just the definite article as mentioned in my first review but also prepositions such as of and in. However, it occurred to me that perhaps for some reasons these authors do not want to reduce the length of the manuscript. Thus, since my editing might be in vain I stopped most editing half way through the manuscript. If this notion is incorrect the authors can follow my examples of the first half to edit the 2nd half accordingly.

Author response – We thank Dr. Hudson for his conscientious editing of the manuscript. As below, we have considered all suggestions, and we have adopted most.

L32. Delete the.
L38. Delete the.
L39. And to with. Change are used to infer to imply.
L40. Delete respectively. The two sets are in order.
L43. Delete the twice.
L44. Delete the. Delete in the. Change changes to variations. Move aerosol in front of variations.
L45. In to within.
L46. From to between. To to and.
L47. Delete the.
L52-3. Change and lower during to than in.
L53. Delete of. Particles singular. Move transport after particle.
L54. Delete the. Delete 2nd of. Move focus after chemistry.
L55. Delete in the. Move Arctic in front of research. Delete the. Add time to spring. Delete period. Delete from. Move transition after summer.
L56. Delete 1st 2 thes. Opportunity plural. Delete in. move changes after chemistry.
L57. Delete last the.
L59. Delete of the. Move Arctic on front of warming.
L72. Change aerosols to particles.
L73. Delete sources. Delete for. Move of after nuclei.
L74. Drop s of towards.
L76. Delete much. Change sometimes to often.
Author response – L32.–L76.: most of the changes have been made.

L82. Explain surface-active. Does this mean hygroscopic?

Author response – Components that lower the surface tension. Details are in the reference.

L83. Delete 1st the.
L84-85. Move summertime and Arctic in front of environment. Delete of the.
L85. Particles singular and move in front of concentrations. Delete of. Above to larger than.
L87. Delete nm.
L89. Delete much.
L91. Period after parentheses. And to Moreover.
L92. At to to. ing to e.
L96. Delete made.
L97. Delete that.
L98. Delete the. Delete in the. Move CDNC after model.
L99. Delete will. Above to for.
L100. Insert greater than before 10. Delete in. move increases after CCN. Delete of the. Change phere to pheric. Move cooling after atmospheric. Change the to This.
L101. Move threshold after concentration.

L101-2. Change and it is noted that the value of 10 cm-3 to although this. Insert a after not. Insert limit after universal.

L103. Delete 2nd the. Add time to summer and move in front of microphysics. Clouds singular and move in front of microphysics.

L107. Insert Yum and Hudson, 2001 and Wylie and Hudson, 2002.

L108. Delete 1st the. Add time to spring.

L109. Aerosols to particles.

L111. Delete of. Aerosols singular and move in front of observations.

L112. Delete during. Add time to summer and move in front of Arctic.

L113. Delete of. Aerosols singular and move in front of measurements. Add time to summer. Move Arctic in front of clouds.

L114. Period after forcing. Insert They to begin next sentence. ing to ed.

L115. Delete about. Move forcing from these plumes after maximum. Half is approximate.

L117. Among to between. Insert and after coupled. Move uncoupled in front of to. delete versus those. Period after surface. Change but to They. Delete of the. Move observations after microphysics. Insert their after and.

L119. Characterization plural. Was to were.

L119-20. Move June in front of low. Delete in.

L121. Delete the. Delete in the. Tops singular. Move cloud top in front of CDNC.

L121-2. Remove quotes. Move aerosols to end of sentence.

L122. Delete the twice. Bases singular. Insert cloud in front of base.

L123. Delete does. Add s to influence. Delete the.

L124. Delete the twice. Delete in. add time to summer. Cloud plural and move after Arctic.

L127. Aerosols and clouds singular and move in front of observations. Remove of.

L128. The to this.

L153. Delete are.

L157. The to these.

L162. The to these.

L165. Add s to detect. Move detects particles right after that. Uses to using. Delete of. Move scattering after light. Delete to.

L167. Delete a reduced pressure of.

L168. Last The to This.

L169. Delete of. Move measurement in front of stability.

L170. Delete the. Delete of. Particles singular. Move hygroscopicity to end of sentence.

L171. Using to with.

L173. Period after parentheses. Delete and.

L176. Dimensions to dimensional. Move two dimensional in front of Cloud. Delete in. sized from about to between. To to and.

L177. Using to with. Delete For. Delete present. The to this.

L177-8. Move this study to the end of this sentence. insert from after phase.

L179. Period after parentheses. Delete and.

L182. Delete The.

L183. Use CO.

L184. Delete at. Move 150nm in front of excitation. The to This.

L191. Measured to done.

L192. The to This.

L198. Semicolon to period. Insert However before the.

L199. Insert or below after in.

L200. Are to was.
L201. Are to was. Is to was.
L205. Delete of.
L206. Delete the.
L207. Delete of. Particles singular. Move transmission after particle. Approximately to near.
L211. Delete at.
L211-2. Move exhaust tube in front of flow.
L212. Move flow after tube. Delete the four times. Delete of twice. Delete at. Delete allowed. Delete last flow. Delete ly.
L213. Move flow after intake. Delete at the. Period after TAS. Delete and.
L216. Delete 1st the. Delete of the. Move aircraft in front of forward.
L217. Change lowered to reduced. 1st the to this.
L220. Delete to. Move Analysis in front of Approach.
L223. Beginning to between. Delete ending July. The to These.
L224. Delete relatively. Delete the. Relatively and distinct are opposites.
L225. Delete The.
L226. Change calm to light. Change varying to variable. Insert the before south. Delete to north.
L233. Period after parentheses. Then insert This was. Delete in part. Possibly is enough of a hedge.
L237. The to this.
L239. Delete of. Flying to legs.
L240. Above to altitude.
L241. Delete the surface. Surface here must me sea level.
L245. Delete surrounding. Period after surfaces. Change as well as to Furthermore. Delete because. Insert was after 2. Insert by after marked.
L246. Put Fig. 2 in front of panel. Delete in.
L247. Flight plural. Delete plans were. Change towards sampling to on.
L251. Change greater to larger. Delete for.
L252. Delete The.
L253. Over to between. Dash to and. Delete data, which are.
L253-4. Move Fig. S3 in front of example.
L254. Change shown in to and. Particle singular and move in front of number. Delete of. Delete 1st nm.
L258. In to within. Move study in front of area. Delete of. Move Within the study area to beginning of sentence. Delete when they. Change ideally to mostly. Ascending to ascents.
L259. Or to and. Descending to descents. Delete through them. Delete the. Base plural. Delete of. Clouds singular. Move bases after cloud.
L260-1. Move only liquid phase clouds after μm.
L262. With the caveat to except.
L264. May to might.
L265. Period to whereas.
L266. Move July 7 in front of stratocumulus. Delete sampled on. But to though.
L270. Insert and when after where. Change was clear and achievable to could be observed. Clear is a poor word choice to describe cloud base. Change semicolon to period.
L272. In to within. Comma to and.
L272-3. Move in flight in front of indistinguishable.
L273. Period after parentheses. And to Thus,. The to such. Cloud plural.

L278. Are to were.

L281-2 and elsewhere. Points is not the best word choice unless you are referring to elements of a figure. Sections or segments are alternatives. Or just refer to data without another word.

L282. Is to was. The to These.

L288. Delete the.

L289. Move valid in front of in-cloud. Delete considered. Delete inside of cloud.

L290. Thermodynamic plural. Delete measurements, it is used twice in this sentence. Delete the. In to within.

L293. On to upon. Insert Hudson and Frisbie [1991] and Hallett and Christensen [1984].

L303. Delete on July 7 sampled. Delete 1st the.

L304. Change and the to while.

L305. Period after altitude. Insert These before features. Change common to the to characterize. Move formation after cloud. Delete of. Change and indicating to so.

L306. Delete cloud. Delete in air rising. Insert below after from.

L308. Delete 1st the.

L309. Delete the. In to Within.

L313. Delete 3rd the. Insert Hudson [1993] at end of sentence. Change The to Thus.

L315. Delete 2nd the.

L317. Insert corresponding before N5.

L318. Delete the. Move closure after concentration (singular). Delete of.

L320. Change down to about to as small as.

L320-1. Move based on maximum CDNC to beginning of sentence.

L321. Delete the. Delete of. Move particles after sulphate.

L322. Change the to that. Delete of the. Clouds singular and move in front of bases.

L324. At to of.

L325. And to while.

L326. Change but there are to except that. Insert is before more. Insert broken after more.

L326-7. Move the July 17 profile right after except that.

L327. Delete what is left of this line.

L328. Delete adiabatic lifting. Insert lower LWC before intervals. Delete with decreasing LWC. Change associated with to due to.

L329. Change the to cloud. Delete of the stratocumulus. Profile plural. Last the to this.

L330. Move LWC in front of peak. Delete in the. Change below to from.

L331. Move CO in front of increase. Delete in. insert at before about.

L332. Move the in front of erosion. Change was to went. In to into. Change case to cloud or clouds.

Author response – L83.-L332: most of the changes have been made.

L333. What aerosol increase above cloud? Aerosol decreases at many levels. Delete the.

Author response – The N100 and CCN, shown in Fig. 3c are higher above cloud than below cloud. We have re-written it to say "indicating that the higher concentrations of N50-100, N100 and CCN above cloud relative to below cloud did not enhance the CDNC"

L334. This is true at a greater altitude range.

Author response – "about" changed to "at least"

L335. This is not shown in the figure. Delete last the.

Author response – N50 is derived from the sum of N50-100 and N100, both of which are shown in the figure. Last "the" deleted.

L336. Larger to higher. Of to between.
L340. 1st the to a.
L342. Last the to this.
L343. Period after study. Delete and. Insert that before BB.
L347. Add ed to reach.
L348. Period after layer. Insert This is.
L349. Reduced to lower. Increased to higher.
L350. Semicolon to comma. Insert which is. Increase to higher. Delete in. insert concentrations above than after aerosol. Delete between. The to this. Delete and.

Author response - L336.-L350.: most of the changes have been made.

L351. Delete above the layer. N50 is apparently not shown in the graph! I do not see these numbers.

Author response – They are derived from N50-100 added to N100.

L354. Delete sized. Delete 1st nm. Insert diameter after nm.
L355. Insert comma which after CDNC. Down to about to as small as.
L356. Reduced to lower.
L357. Insert below cloud after of. And to with. Above cloud CCN does not show this.
L358. Delete case of a. delete in.
L358-9. Move LWC in front of variations.

Author response - L354.-L358-359.: most of the changes have been made.

L359. Delete the. Suggests to suggest. Looks higher than 49.

Author response – Changes made. Because it is narrower, the lower CDNC at the boundaries have a larger influence on the average.

L361. The to These.

Author response – Changed.

L363. I do not see this in the figure.

Author response – N50 is derived from the sum of N50-100 and N100.

L365. Pre to below.

Author response – "pre" is correct.

L365-7. It is problematic to get valid aerosol measurements of any kind in the narrow layer between these clouds.
L366. 44?
L367. 52?, seems higher than 34. Seem higher than 66. Seems higher than 35.
L369. For the lower cloud layer yes, but not so sure about the other two.

Author response – Estimating the aerosol for such layers is problematic, and we make a statement to this effect regarding this case.  Again the N50, which is quoted in the text is derived from the sum of N50-100 and N100, which we think explains the confusion over the N50 number concentrations.

L372. Delete 2nd July.
L373. Delete and July.

Author response – Second change made.

L377. Looks much higher in Fig. 4a.

Author response – The mean is influenced by many low points.

L378. Apparently below cloud is to the right? 16:45 to 17:09 and beyond? This needs to be stated. N100 appears to be 2. 0.6% ammonium sulfate is 40 nm diameter. 100 µm ammonium sulfate is 0.1%.

Author response – The cloud (CDNC represented by blue points) is to the left. Aerosol is to the right, and the wind direction, as indicated, is right to left.  The CCN at 0.6% are higher than the CDNC except at 130 m where the CDNC reaches 12/cc.

L385. Explain these 7 samples. This is not obvious from the figure.

Author response – Explanation added by re-writing the segment as "Seven samples were identified over the period 16:06-16:29 based on the LWC above 0.01 g m-3. The CDNC are overall higher than on July 5 with sample averages ranging from 4 cm-3 to 13 cm-3; the one-second CDNC are as high as 34 cm-3 and the mean VMD (not shown) range from 19.6 um to 22.8 um."

L389. Where is N100 this low? To the right and left sides of Fig. 4b N100 is 50 or more! Lower values are seen within cloud. All measurements where altitude is shown seem to be within cloud and thus invalid for aerosol measurements.

Author response – As stated in the text, "In the air nearly free of cloud and below 120 m".

L390-1. Delete due to instrument problems.

Author response – deleted.

L391-2. These are not the numbers that appear on Fig. 4b.

Author response – The results have been checked several times. They are correct based on our defined approach.

L400. Delete from. East to easterly. Delete to west.
L404. The to This.
L414. The to These.
L418. The to these.
L427. The to these.
L467. The to these.
L533. Insert S after %.
L572. Reach above to exceed.
L578. Insert that before 1st the.

Author response – L400.-L578.: appropriate changes have been made.

[revised manuscript text omitted]